# The *Candida glabrata* Upc2A transcription factor is a global regulator of antifungal drug resistance pathways

Bao Gia Vu[1], Mark A. Stamnes[1], Yu Li[2], P. David Rogers[2¤], W. Scott Moye-Rowley[1]*

1 Department of Molecular Physiology and Biophysics, Carver College of Medicine, University of Iowa, Iowa City, Iowa, United States of America, 2 Department of Clinical Pharmacy and Translational Science, University of Tennessee Health Science Center, Memphis, Tennessee, United States of America

¤ Current address: Department of Pharmaceutical Sciences, St. Jude Children's Hospital, Memphis, Tennessee, United States of America

* scott-moye-rowley@uiowa.edu

**Data Availability Statement:** Large datasets are included in this work. They are available as GEO reference Series: GSE182516.

## Abstract

The most commonly used antifungal drugs are the azole compounds, which interfere with biosynthesis of the fungal-specific sterol: ergosterol. The pathogenic yeast *Candida glabrata* commonly acquires resistance to azole drugs like fluconazole via mutations in a gene encoding a transcription factor called *PDR1*. These *PDR1* mutations lead to overproduction of drug transporter proteins like the ATP-binding cassette transporter Cdr1. In other *Candida* species, mutant forms of a transcription factor called Upc2 are associated with azole resistance, owing to the important role of this protein in control of expression of genes encoding enzymes involved in the ergosterol biosynthetic pathway. Recently, the *C. glabrata* Upc2A factor was demonstrated to be required for normal azole resistance, even in the presence of a hyperactive mutant form of *PDR1*. Using genome-scale approaches, we define the network of genes bound and regulated by Upc2A. By analogy to a previously described hyperactive *UPC2* mutation found in *Saccharomyces cerevisiae*, we generated a similar form of Upc2A in *C. glabrata* called G898D Upc2A. Analysis of Upc2A genomic binding sites demonstrated that wild-type Upc2A binding to target genes was strongly induced by fluconazole while G898D Upc2A bound similarly, irrespective of drug treatment. Transcriptomic analyses revealed that, in addition to the well-described *ERG* genes, a large group of genes encoding components of the translational apparatus along with membrane proteins were responsive to Upc2A. These Upc2A-regulated membrane protein-encoding genes are often targets of the Pdr1 transcription factor, demonstrating the high degree of overlap between these two regulatory networks. Finally, we provide evidence that Upc2A impacts the Pdr1-Cdr1 system and also modulates resistance to caspofungin. These studies provide a new perspective of Upc2A as a master regulator of lipid and membrane protein biosynthesis.

**Funding:** This work was supported by NIAID
R01AI152494 (WSM) and NIAID R01AI131620
(PDR). The funders had no role in study design,
data collection and analysis, decision to publish, or
preparation of the manuscript.

**Competing interests:** The authors have declared
that no competing interests exist.

## Author summary

In the pathogenic yeast *Candida glabrata*, expression of the genes encoding enzymes in
the ergosterol biosynthetic pathway is controlled by the transcription factor Upc2A. *C.
glabrata* has a low intrinsic susceptibility to azole therapy and acquires fluconazole resis-
tance at high frequency. These azole resistant mutants typically contain substitution muta-
tions in a gene encoding the transcription factor Pdr1. Pdr1 does not appear to regulate
ergosterol genes and instead induces expression of genes encoding drug transport pro-
teins like *CDR1*. Here we establish that extensive overlap exists between the regulatory
networks defined by Upc2A and Pdr1. Genomic approaches are used to describe the hun-
dreds of genes regulated by Upc2A that far exceed the well-described impact of this factor
on genes involved in ergosterol biosynthesis. The overlap between Upc2A and Pdr1 is pri-
marily described by co-regulation of genes encoding membrane transporters like *CDR1*.
We provide evidence that Upc2A impacts the transcriptional control of the *FKS1* gene,
producing a target of a second major class of antifungal drugs, the echinocandins. Our
data are consistent with Upc2A playing a role as a master regulator coordinating the syn-
thesis of membrane structural components, both at the level of lipids and proteins, to pro-
duce properly functional biological membranes.

## Introduction

An almost inescapable problem for chemotherapy of microbes is the development of resis-
tance. This problem is especially acute in the case of pathogenic fungi for which only 3 differ-
ent drug classes exist for use in treatment of infections (reviewed in [1,2]). The most
commonly used drug class is the azole compounds, chief among these is the well-tolerated flu-
conazole (reviewed in [3]). Fluconazole targets ergosterol biosynthesis and has been used
extensively since the 1980s but this wide usage has led to the development of resistant organ-
isms (recently discussed in [4]). The prevalence of fluconazole as an anti-Candidal therapy has
likely contributed to the changing epidemiology of candidemias with the frequency of these
fungal infections being increasingly associated with *Candida glabrata*; an increase that corre-
lates with the introduction of fluconazole as an antifungal drug [5].

C. glabrata exhibits two features that complicate its control by fluconazole. First, this patho-
gen has a high intrinsic resistance to fluconazole [6]. Second, high level resistant isolates easily
arise that contain gain-of-function (GOF) mutations in a transcription factor-encoding gene
called *PDR1* [7–9]. The GOF *PDR1* alleles exhibit high levels of target gene expression and
drive robust fluconazole resistance primarily through induction of expression of the ATP-
binding cassette transporter-encoding gene *CDR1* [10,11].

The primary species associated with candidemias is *Candida albicans* (Ca) which can also
acquire fluconazole resistance (recently discussed in [12]. Interestingly, the range of genes in
which mutations are observed to associate with fluconazole resistance in *C. albicans* is much
wider than in *C. glabrata*. Along with mutant forms of the genes encoding the well-described
transcription factors CaTac1 and CaMrr1 [13,14], two additional genes in which fluconazole
resistant alleles can emerge in *C. albicans* are Ca*ERG11* [15], that encodes the enzymatic target
of azole drugs, and CaUpc2, the primary transcriptional activator of Ca*ERG11* and other
ergosterol biosynthetic pathway genes [16]. Mutations in the cognate genes for these proteins
have not been found in *C. glabrata*.

Two important observations have recently linked *C. glabrata* Pdr1 with the ergosterol bio-
synthetic pathway in this yeast. First, loss of *UPC2A* (*C. glabrata* homologue of *C. albicans*

*UPC2*) was sufficient to strongly reduce fluconazole resistance of a GOF *PDR1* mutant allele [17]. Second, genetic means of reducing the flux through the ergosterol pathway led to induction of the Pdr pathway, including *PDR1* and *CDR1*, in a Upc2A-dependent manner [18]. Together, these data indicated that fluconazole resistance in *C. glabrata* was likely to involve coordination of the Pdr1- and Upc2A-dependent transcriptional circuits.

Given that Upc2A interfaced with the *PDR1* and *CDR1* promoters, we wanted to determine the full spectrum of genes bound and regulated by this factor. This was accomplished using chromatin immunoprecipitation coupled with Next Generation Sequencing (ChIP-seq). We also performed RNA-seq studies to identify the Upc2A-dependent transcriptome. Using the strong sequence conservation between *Saccharomyces cerevisiae* (Sc) Upc2 (ScUpc2) and Upc2A, we constructed a GOF form of Upc2A in *C. glabrata* based on an allele described for its *S. cerevisiae* homologue [19]. This mutant Upc2A drove elevated fluconazole resistance and behaved like the hyperactive *S. cerevisiae* factor. ChIP-seq data indicated Upc2A bound to roughly 1000 genes and that this binding was highly induced by fluconazole. Comparison of the genes bound by Upc2A with those we previously found to be associated with Pdr1 indicated a high degree of overlap between these two target gene suites. Transcription of the *FKS1* gene, encoding a β-glucan synthase protein was also found to be responsive to Upc2A, consistent with *upc2AΔ* strains being hypersensitive to caspofungin which is thought to act as a β-glucan synthase inhibitor (reviewed in [20]). Our data provide a new view of the global importance of Upc2A-mediated transcriptional activation as extending far beyond its well-appreciated control of *ERG* gene expression. Upc2A appears to serve as a key coordinator of the biosynthesis of lipids and proteins that are destined to function in this membrane environment.

## Results

### A gain-of-function form of Upc2A confers elevated fluconazole resistance

Fluconazole resistance is most commonly caused in *C. glabrata* by substitution mutations within the *PDR1* gene (See [6,21] for a review). These mutant transcription factors exhibit enhanced target gene expression when compared to the wild-type allele. Although mutations in the *UPC2* gene in both *Saccharomyces cerevisiae* and *Candida albicans* have been isolated that drive fluconazole resistance [22–24], there are no gain-of-function (GOF) forms currently known for *UPC2A*. To determine if a GOF allele of *UPC2A* could be produced, we constructed an allele based on analogy with a mutation first found in Sc*UPC2* which caused the enhanced function of this transcriptional regulator [25]. The relevant mutation (*upc2-1*) [19] is a change of a glycine to an aspartate residue located at position (G888D) in the carboxy-terminus of ScUpc2. Alignment of *C. glabrata* Upc2A and ScUpc2 indicated that G898 was the analogous position in Upc2A. This residue was replaced with an aspartate to form the G898D *UPC2A* form of the gene. The resulting mutant allele was tagged with a 3X hemagglutinin (3X HA) epitope at its amino terminus as we have previously done for the wild-type *UPC2A* gene and both these forms of *UPC2A* were inserted in place of the native *UPC2A* locus in an otherwise wild-type *C. glabrata* strain. These tagged strains (See Table 1) were then grown to mid-log phase along with isogenic wild-type and *upc2AΔ* cells. Serial dilutions of each culture were placed on rich medium containing the indicated concentrations of fluconazole (Fig 1A).

Introduction of the G898D mutation into *UPC2A* led to the resulting factor exhibiting elevated fluconazole resistance when compared to either the tagged or untagged version of the wild-type gene. Loss of *UPC2A* caused a dramatic increase in fluconazole susceptibility. This is consistent with G898D *UPC2A* behaving as a hyperactive transcriptional activator in *C. glabrata* as has previously been seen for G888D Upc2 in *S. cerevisiae* [25].

**Table 1. Strains and relevant genotypes for *C. glabrata* cells used in this study.**

| Name | Parent strain | Genotype |
|------|---------------|----------|
| KKY2001 | CBS138 | *his3Δ*::FRT *leu2Δ*::FRT *trp1Δ*::FRT |
| BVGC3 | KKY2001 | *ERG11*-3X HA::*his3MX6* KKY2001 |
| BVGC138 | BVGC3 | *pdr1Δ*::*loxP* KKY2001 |
| BVGC205 | BVGC138 | *PDR1*::*loxP* KKY2001 |
| BVGC207 | BVGC138 | mSRE *PDR1*::*loxP* KKY2001 |
| BVGC209 | BVGC205 | *PDR1*::*loxP cdr1Δ* KKY2001 |
| BVGC213 | BVGC207 | mSRE *PDR1*::*loxP cdr1Δ* KKY2001 |
| BVGC268 | BVGC213 | mSRE *PDR1*::*loxP cdr1Δ* KKY2001 *LEU2* |
| BVGC59 | BVGC3 | *upc2AΔ* KKY2001 |
| BVGC82 | BVGC59 | *UPC2A*-3X HA-*UPC2A*::*loxP* KKY2001 |
| BVGC84 | BVGC59 | *UPC2A*-3X HA-G898D *UPC2A*::*loxP* KKY2001 |

To characterize the action of G898D Upc2A in control of fluconazole resistance, the expression of a range of different genes involved in this phenotype was examined by RT-qPCR. Our previous experiments have identified Upc2A as an activator of expression of both the ATP-binding cassette transporter-encoding gene *CDR1* and the *PDR1* gene encoding a key transcriptional activator of *CDR1* [26]. Levels of mRNA for the *ERG11* gene, encoding the enzymatic target of fluconazole, as well as for *UPC2A* itself were also evaluated in the presence and absence of azole drug challenge.

The presence of the G898D *UPC2A* gene led to a strong elevation of *ERG11* transcription in the absence of fluconazole but had no significant effect on the other genes (Fig 1B). Treatment with fluconazole elevated transcription of all Upc2A target genes although the elevated *ERG11* mRNA levels seen in the absence of drug were not further induced by fluconazole when G898D Upc2A was present. The presence of fluconazole elevated *PDR1* and *CDR1* by 4- or 7-fold, respectively when the wild-type *UPC2A* gene was present. Introduction of the G898D *UPC2A* allele modestly reduced fluconazole induction for these two genes to 2-fold for *PDR1* and 4-fold for *CDR1*. It is possible that the increased levels of *ERG11* expression caused by the G898D Upc2A protein could influence fluconazole induction of *PDR1* and *CDR1*. *UPC2A* transcription was elevated approximately two-fold by fluconazole, irrespective of the *UPC2A* allele tested.

These same strains were then used to compare expression of their protein products by western analysis with appropriate antibodies. All strains were grown in the absence or presence of fluconazole and whole cell protein extracts prepared. These were analyzed by western blotting using the indicated antibodies (Fig 1C).

The presence of the G899D *UPC2A* allele supported normal fluconazole induction of Cdr1 and showed a modest reduction (~80% of wild-type) in Pdr1 activation. The levels of the wild-type and G898D forms of Upc2A were not detectably different as shown by blotting with the anti-Upc2A polyclonal antiserum. Over the time course of fluconazole challenge (two hours), no differences in the levels of these two forms of Upc2A were seen. These data argue that the increased activation seen for *ERG11* in the presence of the G898D *UPC2A* gene was due to increased function of Upc2A rather than a change in its expression compared to the wild-type factor.

## Global analysis of Upc2A direct transcriptional targets

Having confirmed that the HA-tagged form of both wild-type and G898D Upc2A behaved as expected, we used these forms of Upc2A to carry out chromatin immunoprecipitation coupled

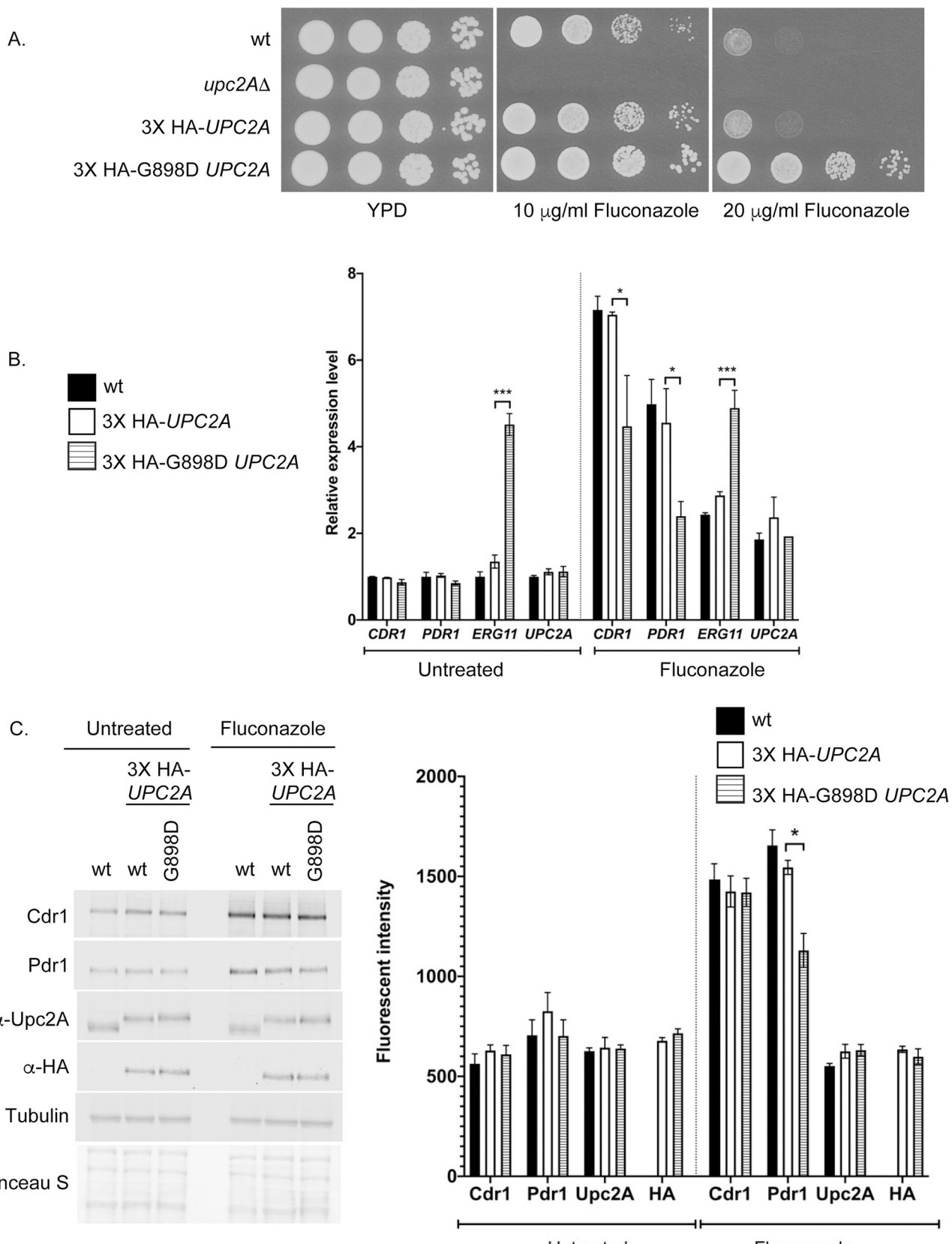

**Fig 1. Characterization of *UPC2A* gain-of-function allele.** A. An isogenic series of strains was prepared that varied in their *UPC2A* allele: wild-type (CBS138), *upc2AΔ*, or strains containing an amino-terminally HA-tagged form of wild-type (3X HA-*UPC2A*) or the G898D form of *UPC2A* (3X HA-G898D *UPC2A*). These strains were grown to mid-log phase and then tested for their resistance to the indicated levels of fluconazole in rich medium. B. G898D Upc2A induced *ERG11* mRNA under basal conditions. The strains described in A were grown in the presence or absence of fluconazole and total RNA prepared. Levels of mRNA were assessed by RT-qPCR. C. Western blot analysis of Upc2A and target gene-encoded proteins. The indicated strains were grown as described previously in the presence or absence of fluconazole (20 µg/ml), whole cell protein extracts prepared and analyzed by western blotting using the antisera listed at the left side. Upc2A was detected using either an anti-Upc2A polyclonal antibody (α-Upc2A) or anti-HA (α-HA). Tubulin was used as a loading control along with Ponceau S staining of the membranes. Quantitation shown in the right hand panel was performed as described in Materials and Methods.

with Next Generation Sequencing (ChIP-seq) to identify the genomic targets of Upc2A. Additionally, we assessed the effect of fluconazole induction on the suite of genes bound by either the wild-type or G898D forms of Upc2A. The strains used above were grown to mid-log phase under control or fluconazole-treated conditions and fixed chromatin prepared and fragmented. Chromatin was immunoprecipitated with anti-HA antibody, purified and analyzed by Illumina sequencing. Reads were mapped and peaks called by use of the MACS2 algorithm [27]. We first analyzed the peaks seen in these 4 different conditions (wild-type +/- fluconazole, mutant G898D +/- fluconazole) by determining the overlap of bound genes between them (Fig 2A).

The Venn diagram shown illustrates the extent of overlap between each different ChIP-seq condition. The largest number of bound genes was seen for the wild-type factor in the presence of fluconazole (>1000 promoters bound). Interestingly, there were 565 of these promoters that could only be bound by Upc2A in the presence of fluconazole while others were bound in multiple different conditions. This large group of Upc2A-bound genes include loci not typically considered as target genes, such as components of the translational machinery, and await experimental analysis to test their functionality.

The next largest group of bound promoters (309) were found to be bound under all 4 different conditions. The third largest class of promoters (85) were only bound by Upc2A under conditions we consider induced (wild-type + fluconazole and G898D Upc2A +/- fluconazole). GO term enrichment analysis of these three different classes of promoters indicated that very different genes were associated with these patterns of Upc2A DNA-binding. The largest class of genes was enriched for components of the translation apparatus (S1A Table). The class of genes bound under all 4 different conditions was most predominantly enriched for proteins associated with the plasma membrane (S1B Table) while the inducible class of Upc2A-bound promoters showed the strongest enrichment for genes involved in fatty acid biosynthesis (S1C Table). While all these genes are Upc2A target loci, they exhibited unique patterns of association with this factor.

We used the software MEME-ChIP to search the peaks associated with binding of wild-type Upc2A in the presence of fluconazole for sequence motifs that were enriched in this collection of binding sites. We chose this condition as it represented the broadest collection of sterol response elements (SREs). We compared the MEME-ChIP output to known binding sites for Upc2 in *S. cerevisiae* (ScUpc2). The SREs in *S. cerevisiae* were also referred to as anaerobic response elements (AR) [28,29]. Here, we refer to the binding elements for Upc2A in *C. glabrata* as SREs on the basis of their similarity to those previously described for their role in ergosterol biosynthetic gene regulation in *S. cerevisiae* [30]. This analysis is shown in Fig 2B.

The AR1b/c elements show the SRE variations that are tolerated by ScUpc2 in *S. cerevisiae*. These ScUpc2 SREs are most closely fit to the right-hand element of the Upc2A SRE predicted by MEME-ChIP. The central CGTA sequence is conserved between *C. glabrata* and *S. cerevisiae*, although the *C. glabrata* element has a nearby conserved element (CACAGA) that shows a relatively constant spacing. It is important to note that all analyses of *S. cerevisiae* Upc2 DNA

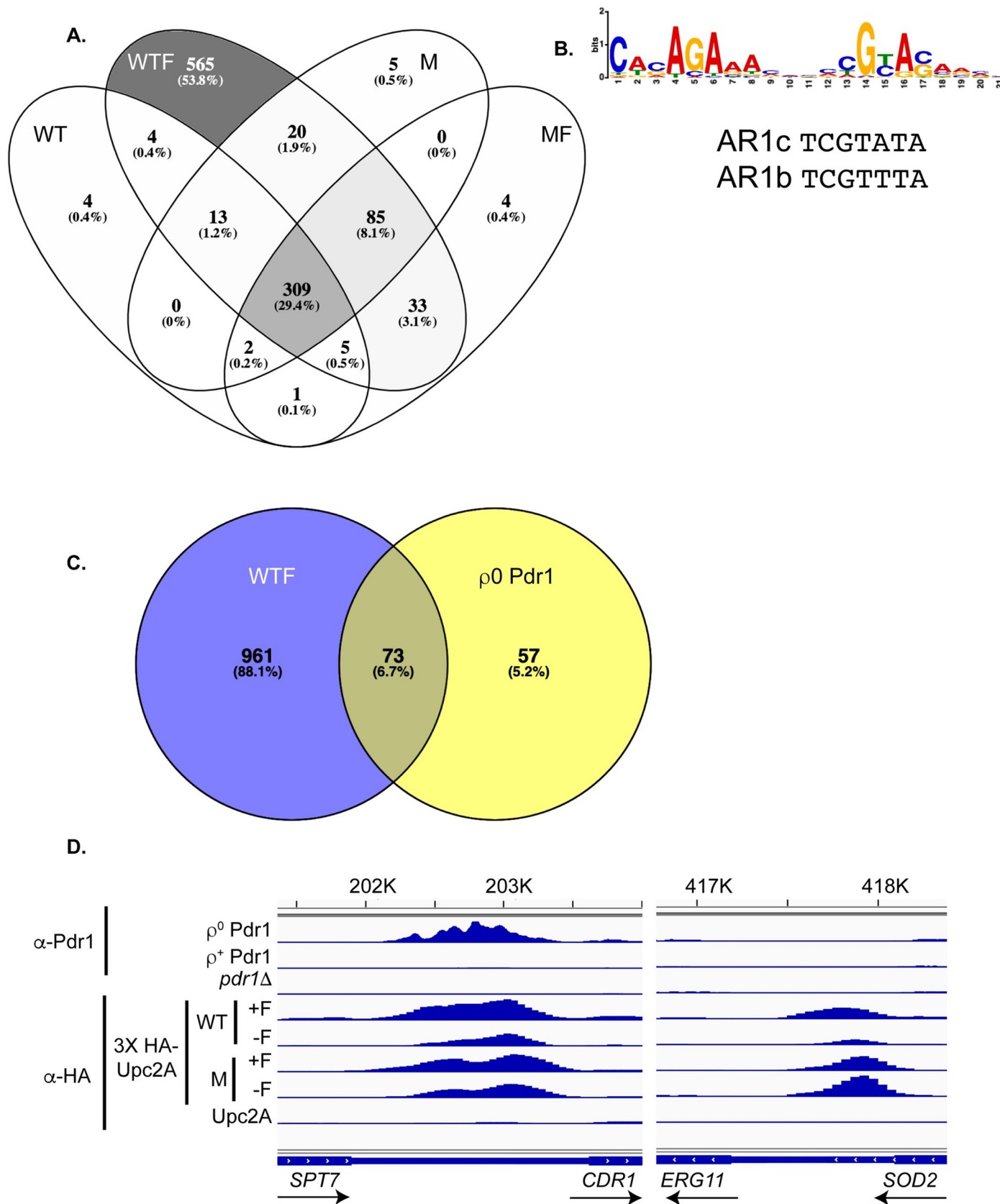

**Fig 2. Chromatin immunoprecipitation-high throughput sequencing (ChIP-seq) analysis of *UPC2A*.** A. 3X HA-tagged forms of either the wild-type (WT) or G898D (M) alleles of *UPC2A* were used for a standard ChIP-seq experiment. Strains were in the presence (WTF, MF) or absence (WT, M) of fluconazole (20 µg/ml), followed by ChIP-seq analysis as described earlier [31]. A Venn diagram showing the overlap of genes detected in each sample is shown with largest number of genes color-coded from dark to light. B. MEME-ChIP analysis of sterol response element (SREs) shared in Upc2A peaks. A logo is shown representing the most commonly enriched element associated with Upc2A ChIP-seq peaks that was detected by MEME-ChIP analysis. AR1b and AR1c show the corresponding Upc2 consensus sites from *Saccharomyces cerevisiae* [29]. C. Venn diagram showing the overlap between binding sites found for fluconazole-stressed 3X HA-*UPC2A* (WTF) and $\rho^0$-induced wild-type Pdr1 [31]. D. Comparison of Upc2A- and Pdr1-binding to the *CDR1* and *ERG11* promoters. Plots of relative ChIP-seq density are shown for ChIP reactions performed with either anti-Pdr1 or α-HA to detect epitope-tagged Upc2A. ChIP in samples treated with fluconazole is denoted +F while the corresponding no drug control is represented by -F. The bottom line is a control ChIP reaction using wild-type cells that lack the HA-epitope tag (Upc2A). *CDR1* is controlled by both factors while *ERG11* only responds to Upc2A.

binding were carried out before the availability of global approaches like ChIP-seq which will impact predictions of consensus elements as a more limited repertoire of regulated genes was considered. We examine the binding of Upc2A to its DNA target sites in detail below.

## Overlap between Upc2A- and Pdr1-controlled genes by ChIP-seq

These data agree with the model proposed earlier for ScUpc2 that Upc2A appeared to accumulate inside the nucleus upon ergosterol limitation where the factor then binds to SRE-containing promoters [22]. Since our previous work indicated overlap between Upc2A and Pdr1 target genes [26], we wanted to examine the degree of overlap between these different regulons. To make this comparison, we examined the shared Upc2A targets in cells treated with fluconazole (WTF) and the promoters bound by Pdr1 in cells lacking the mitochondrial genome ($\rho^0$ cells), which have been previously published [31]. We selected these two conditions for comparison since both involve a signal that activates the wild-type versions of Upc2A and/or Pdr1 (fluconazole or $\rho^0$ cells, respectively) [8,32]. Strikingly, more than 50% of Pdr1 target promoters were also associated with Upc2A binding (Fig 2C). The top 4 GO terms enriched in genes bound by both these $Zn_2Cys_6$-containing transcription factors were associated with transmembrane transport or integral membrane components (S2 Table). As our earlier work had shown that both *PDR1* and *CDR1* were targets of Upc2A along with Pdr1 [18], these new data indicate that the overlap between these two transcriptional circuits extends well beyond the initial two genes.

Two different classes of Upc2A target promoters are shown in Fig 2D. The *ERG11* gene is an example of a locus controlled by Upc2A but not Pdr1. Binding of wild-type Upc2A is represented by the read depth and can be seen to increase in the presence of fluconazole compared to in the absence of the drug. Binding of G898D Upc2A was constant, irrespective of the presence of the drug. Note that when the lack of a change in Upc2A expression is considered (see Fig 1C), these data support the view that the DNA-binding activity of wild-type but not G898D Upc2A is increased by the presence of fluconazole, possibly by an increase in nuclear localization [22].

*CDR1* represents a Upc2A target gene that is also regulated by Pdr1. The bound regions for Pdr1 and Upc2A extensively overlap in the upstream region of *CDR1*. Pdr1 DNA-binding was strongly upregulated in $\rho^0$ cells, likely due in part to the large increase in *PDR1* expression in this background compared to wild-type cells [31]. Upc2A DNA-binding to *CDR1* was regulated in a manner similar to that seen for *ERG11*.

## Transcriptomics of Upc2A-regulated genes

The data above did not take into account a consideration of target gene expression. In order to link regulated Upc2A binding to changes in gene transcription of target genes, we carried out two additional analyses. First, we used software contained within the MACS2 algorithm called BDGdiff [27] that examines ChIP-seq data and identifies peaks that exhibit significant binding

differences in the comparison of data from fluconazole-treated cells versus untreated cells. These binding sites will be considered fluconazole-regulated. Secondly, RNA-seq assays were performed on isogenic wild-type and *upc2AΔ* cells in the presence and absence of fluconazole. RNA-seq data were processed and we focused on genes that had an adjusted P-value of <0.05 and significantly up-regulated by at least two-fold. These data are summarized in Fig 3 and included in S3 Table.

Fig 3A represents the union of all genes that are induced in the presence of fluconazole in either wild-type or *upc2AΔ* cells with genes that exhibited a significant increase in Upc2A DNA-binding when comparing ChIP-seq data from cells grown in the presence of fluconazole versus the absence of drug. Strikingly, only 53 genes required the presence of Upc2A DNA-binding activity to be induced by fluconazole while 274 were induced either in the presence or the absence of the *UPC2A* gene, although all these genes were bound by Upc2A. The majority of fluconazole-induced genes (542) were not dependent on the presence of Upc2A while 64 genes were fluconazole-induced, required the presence of Upc2A but were not bound by this factor.

These data argue that the vast number of fluconazole-induced genes do not depend on the presence of Upc2A. However, there are 117 genes that are only FLC-inducible in the presence of Upc2A. The only GO term enriched in the 64 genes that require the presence of Upc2A for FLC induction (yet are not bound by Upc2A) represent loci involved in ergosterol biosynthesis. These Upc2A-dependent but indirect targets include *ERG4*, *ERG8*, *ERG9*, *ERG24*, *ERG26* and *ERG27*. *ERG8* is in the earlier part of the ergosterol biosynthetic pathway while all other enzymes participate in the conversion of farnesyl pyrophosphate to ergosterol (recently reviewed in [33]).

The simplest class of fluconazole-induced genes are represented by the 53 direct target genes. These genes are both bound by Upc2A and respond to regulation via this factor. GO terms enriched in this set of genes included ergosterol biosynthesis, plasma membrane and cell wall biogenesis, membrane transport, sterol uptake and RNA polymerase II core promoter-binding factors. The genes in this class of enriched genes include *UPC2B* and homologues of *S. cerevisiae NRG2* and *ADR1*. These factors may be involved in regulation of genes that are controlled indirectly by Upc2A. Examples of this class of indirectly Upc2A-regulated genes would include *ERG* genes like *ERG4* mentioned above.

The other two categories of fluconazole-regulated Upc2A-bound genes were either induced in both wild-type and *upc2AΔ* cells (201 genes) or only in the absence of *UPC2A* (73 genes). Only the group of 201 genes showed any significant GO term enrichment with specific groups of genes involved in the response to stress and glycogen catabolism being the top two categories. These genes illustrate a more complicated mode of regulation by Upc2A as they can be fluconazole-regulated in the presence or absence of this factor. Further dissection of the Upc2A regulon is required to illuminate the modes of control of these genes.

The final and largest class of genes found to be bound by Upc2A were less than two-fold induced by fluconazole. GO term analysis of this class of genes indicated that the top enriched categories were involved in translation and the ribosome (60 total). The next highest categories were plasma membrane or amino acid transmembrane transport and represented 74 genes. Together, these data strongly suggest that Upc2A impacts a wide range of cellular process as well as its well-described control of expression of genes involved in the ergosterol biosynthetic pathway.

**Upc2A-responsive genes and the impact of Pdr1.** To examine the expression of Upc2A-responsive genes and the effect of Pdr1 on selected loci, we compared the transcriptional response of a range of genes from the *ERG* pathway with loci that we have previously found to be targets of Pdr1 in *C. glabrata* [31]. This comparison is presented in the form of a heat map (Fig 3B).

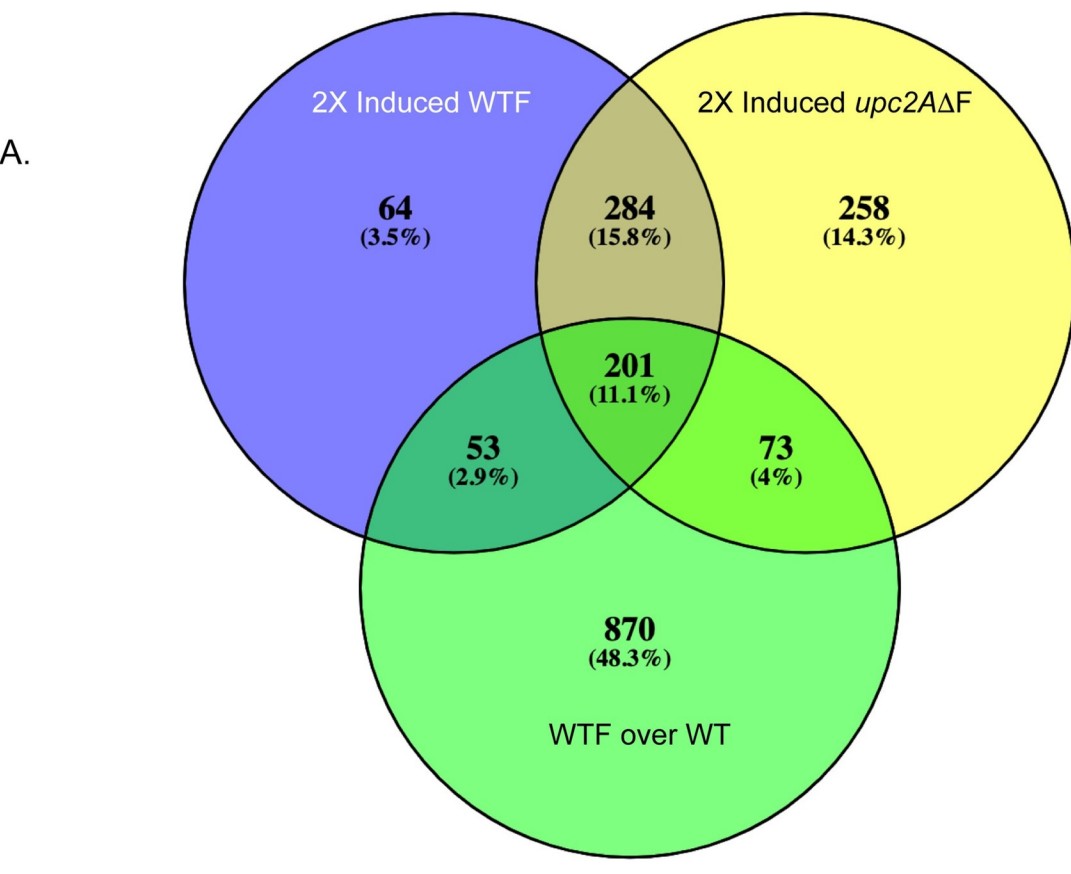

| SRE | Log2 FLC wt | Log2 FLC upc2A | Cg | Sc | Product |
|---|---|---|---|---|---|
| ● CAGL0D05940g | 3.6 | 3.6 | ERG1 | ERG1 | Squalene epoxidase |
| CAGL0L12364g | 0.5 | -1.4 | ERG10 | ERG10 | Acetyl-CoA C-acetyltransferase activity |
| ● CAGL0E04334g | 1.9 | -1.4 | ERG11 | ERG11 | Cytochrome P-450 lanosterol 14-alpha-demethylase |
| ● CAGL0H04081g | 0.3 | -1.6 | ERG13 | ERG13 | 3-hydroxy-3-methylglutaryl coenzyme A synthase |
| ● CAGL0L10714g | 2.2 | -1.3 | ERG2 | ERG2 | C-8 sterol isomerase |
| CAGL0L00319g | 0.4 | -0.9 | ERG20 | ERG20 | Putative farnesyl pyrophosphate synthetase |
| ● CAGL0F01793g | 2.6 | 0.1 | ERG3 | ERG3 | Delta 5,6 sterol desaturase |
| CAGL0A00429g | 1.6 | -2.0 | ERG4 | ERG4 | C24 sterol reductase |
| ● CAGL0M07656g | 1.2 | -0.9 | ERG5 | ERG5 | C22 sterol desaturase |
| ● CAGL0H04653g | 0.8 | -1.4 | ERG6 | ERG6 | C24 sterol methyltransferase |
| CAGL0F03993g | 0.9 | 0.7 | ERG8 | ERG8 | Phosphomevalonate kinase |
| CAGL0M07095g | 1.1 | -0.3 | ERG9 | ERG9 | Squalene synthase |
| ● CAGL0L11506g | 0.1 | -1.9 | HMG1 | HMG1 | Hydroxymethylglutaryl-CoA reductase |
| ● CAGL0M01760g | 3.6 | 2.2 | CDR1 | PDR5 | Multidrug transporter of ATP-binding cassette ABC superfamily |
| ● CAGL0F02717g | 2.7 | 0.7 | PDH1 | PDR15 | Multidrug ABC transporter |
| ● CAGL0A00451g | 2.2 | 1.8 | PDR1 | PDR1 | Zinc finger transcription factor |
| ● CAGL0I04862g | 1.4 | 0.4 | SNQ2 | SNQ2 | Multidrug ABC transporter |
| ● CAGL0K00715g | 3.4 | 4.4 | RTA1 | RTA1 | 7 transmembrane domain protein |
| ● CAGL0G08624g | 0.9 | 2.1 | QDR2 | QDR1 | Major Facilitator Superfamily transporter |
| ● CAGL0J07436g | -0.2 | -1.7 | PDR16 | PDR16 | Phosphatidylinositol transfer protein |
| ● CAGL0L10142g | -1.4 | -2.1 | RSB1 | RSB1 | 7 transmembrane domain protein |

-4  -2  -1  1  2  4

**Fig 3. RNA-seq analysis of fluconazole-induced genes in wild-type and *upc2AΔ* strains.** A. Venn diagrams illustrating the union of genes that are at least two fold-induced in wild-type cells treated with FLC (50 μg/ml) compared to untreated (2X Induced WTF), two fold-induced isogenic *upc2AΔ* cells treated with FLC compared to untreated *upc2AΔ* (2X Induced *upc2AΔ*F) along with genes containing a Upc2A SRE that is exhibits fluconazole-inducible Upc2A binding compared to untreated samples (WTF over WT). B. Heat map of representative fluconazole induced genes is shown. The values refer to the log2 score of the ratio of RPKM of fluconazole-treated over untreated samples. The scale for the heat map is indicated at the bottom and the presence of a SRE is denoted by the solid dot in the left hand column.

The majority of *ERG* genes showed at least two-fold induction by fluconazole in wild-type cells as long as these genes corresponded to steps later in the ergosterol biosynthetic pathway. *ERG10*, *ERG13*, *HMG1* and *ERG20*, which all encode early steps in ergosterol biosynthesis, were not influenced by fluconazole challenge in wild-type cells although expression of these genes was strongly depressed under these same conditions in the absence of *UPC2A*. Genes encoding enzymes that function later in ergosterol biosynthesis (like *ERG11*) were induced by fluconazole at least two-fold in wild-type cells but depressed by at least two-fold in a *upc2AΔ* background.

There were two *ERG* genes that were exceptions to these general trends of regulation. *ERG1*, one of the genes showing the best induction by fluconazole in wild-type cells and most prominent peak of Upc2A ChIP-seq density (See S3 Table), was similarly drug induced in both wild-type and *upc2AΔ* cells. *ERG8*, encoding an enzyme that functions early in the ergosterol pathway, was also similarly fluconazole-induced in both gene backgrounds, irrespective of the presence of Upc2A. Strikingly, *ERG8* was not seen to contain a detectable ChIP-seq peak for Upc2A binding. We interpret these complex responses to fluconazole treatment in *C. glabrata* as evidence for a multifactorial transcriptional network regulating *ERG* gene expression in which Upc2A participates as both a direct (later pathway genes) and indirect (early pathway genes) regulator.

Pdr1-regulated genes that were also associated with SREs exhibited less dependence on the presence of Upc2A for fluconazole induction than seen for most *ERG* genes. *CDR1* fluconazole induction was reduced upon loss of Upc2A but *PDR1* was similarly induced irrespective of the *UPC2A* background (Fig 3B). *SNQ2* and *PDH1* were reduced to approximately 50% of their normal drug-induced levels in the *upc2AΔ* strain while *QDR2* and *RTA1* showed higher levels of fluconazole induction in the same strain. Two other Pdr1 target genes (*PDR16*, *RSB1*) were repressed by the presence of fluconazole in wild-type cells and their expression lowered further when *upc2AΔ* cells were treated with fluconazole.

**Limited range of genes regulated by G898D Upc2A.** We also used RNA-seq to compare the gene expression profile of wild-type cells to an isogenic G898D *UPC2A* strain. These strains were grown to mid-log phase in the absence of fluconazole and then standard RNA-seq was carried out to determine the effects of this form of Upc2A on the transcriptome (Table 2).

The presence of the GOF form of *UPC2A* caused relatively small changes in gene transcription. There were only 11 genes observed to be elevated at least 1.4-fold. A striking feature shared by these genes was that nine of eleven encoded products that were involved in the biosynthesis of ergosterol. Five of these 9 genes also contained SREs. As we have seen for *ERG11* (Fig 1B and 1C), the presence of the G898D Upc2A protein led to enhanced expression of multiple genes involved in ergosterol biosynthesis. This coordinate up-regulation is likely responsible for the observed increase in fluconazole resistance caused by this allele.

## Identification of functional SREs in Upc2A target genes

To confirm that the SREs present in the direct Upc2A target genes were required for normal in vivo functions, we both mapped the location of several SREs and prepared mutant versions of these sites that could not normally interact with Upc2A. DNase I protection assays with a

**Table 2. Genes induced by log2 of 0.5 or greater in the presence of the G898D *UPC2A*.** RNA-seq was used to compare the levels of transcripts produced in the presence of the G898D *UPC2A* allele and the ratio of these mRNA levels/those seen in the presence of wild-type *UPC2A* presented as log2 Fold Change.

| Gene | baseMean | log2FoldChange | padj | SRE | Gene | Sc homologue | Function |
|---|---|---|---|---|---|---|---|
| CAGL0F08965g | 39195.9 | 0.87 | 6.17E-11 | No | | MSC7 | Orthologs have role in reciprocal meiotic recombination |
| CAGL0L10714g | 38381.7 | 0.75 | 3.60E-05 | Yes | ERG2 | ERG2 | C-8 sterol isomerase |
| CAGL0J03916g | 9198.9 | 0.71 | 1.54E-06 | No | HES1 | HES1 | Orthologs have oxysterol binding and sterol transport |
| CAGL0K03927g | 10420.4 | 0.68 | 0.00029389 | No | ERG29 | ERG29 | Orthologs have role in ergosterol biosynthetic process |
| CAGL0M07656g | 50162.8 | 0.67 | 0.00075178 | Yes | ERG5 | ERG5 | Putative C22 sterol desaturase |
| CAGL0H04081g | 49335.7 | 0.67 | 0.00075178 | Yes | ERG13 | ERG13 | 3-hydroxy-3-methylglutaryl coenzyme A synthase |
| CAGL0L03828g | 8696.4 | 0.6 | 0.00493476 | Yes | CYB5 | CYB5 | Orthologs have, role in ergosterol biosynthetic process |
| CAGL0A00429g | 36653.4 | 0.59 | 0.00237241 | No | ERG4 | ERG4 | Putative C24 sterol reductase |
| CAGL0K04455g | 236.3 | 0.52 | 0.04177062 | No | | SPR3 | Orthologs have role in ascospore formation |
| CAGL0H04653g | 119991.9 | 0.5 | 0.0436892 | Yes | ERG6 | ERG6 | C24 sterol methyltransferase |
| CAGL0L12364g | 37830.9 | 0.5 | 0.01611664 | No | ERG10 | ERG10 | Orthologs have role in ergosterol biosynthetic process |

recombinant form of the Upc2A DNA-binding domain were used to locate each SRE at nucleotide resolution. Radioactive probes were prepared from SREs contained in the *ERG1*, *CDR1* and *PDR1* promoters. We selected *ERG1* for analysis as the promoter for this gene exhibited some of the strongest binding detected by ChIP-seq These probes were used in a DNase I protection mapping experiment to locate the bounds of the region protected by Upc2A from nuclease digestion. The DNase I ladders were electrophoresed in parallel with chemical sequencing reactions on the same probe in order to locate the SRE. These data are shown in Fig 4.

The *ERG1* promoter, which contains an everted pair of SREs (Fig 4A), showed the largest protected region of DNA and a strong DNase I hypersensitive site located immediately upstream of the SRE (Fig 4B). The *CDR1* SRE exhibited two DNase I hypersensitive sites linked to Upc2A binding while the *PDR1* SRE showed a clear protected region but no associated hypersensitive site.

Now that we could localize the SREs in each of these promoters to a relatively small segment of DNA, we mutagenized each to confirm its requirement for in vitro binding. To confirm that the predicted SREs were key for Upc2A binding, we used an electrophoretic mobility shift assay (EMSA) and prepared wild-type and mutant probes containing each SRE. Each probe was incubated with Upc2A and then resolved using nondenaturing electrophoresis. Bound and unbound probe was detected using a biotin moiety attached to the end of each probe. The results of this assay are shown in Fig 4C.

The wild-type *ERG1* probe produced two different species of protein:DNA complex, possibly corresponding to either one or two binding sites being occupied with Upc2A. The mutant form of this SRE blocked formation of both complexes. *CDR1* and *PDR1* probes both formed primarily a single size of complex that was greatly diminished when the mutant SRE probe was used. These data argue that the SREs indicated in Fig 4A are likely required for Upc2A binding to each promoter.

## Phenotypes caused by loss of SRE function

To validate the importance of the SREs identified in *ERG1*, *CDR1* and *PDR1*, we prepared versions of these promoters that contained the mutations shown to block in vitro binding of Upc2A. These DNA-binding defective SREs (mSRE) were first introduced, along with their wild-type promoter, into a *lacZ* fusion plasmid to allow comparison of the expression supported by wild-type promoters to those that lacking Upc2A binding. These plasmids were

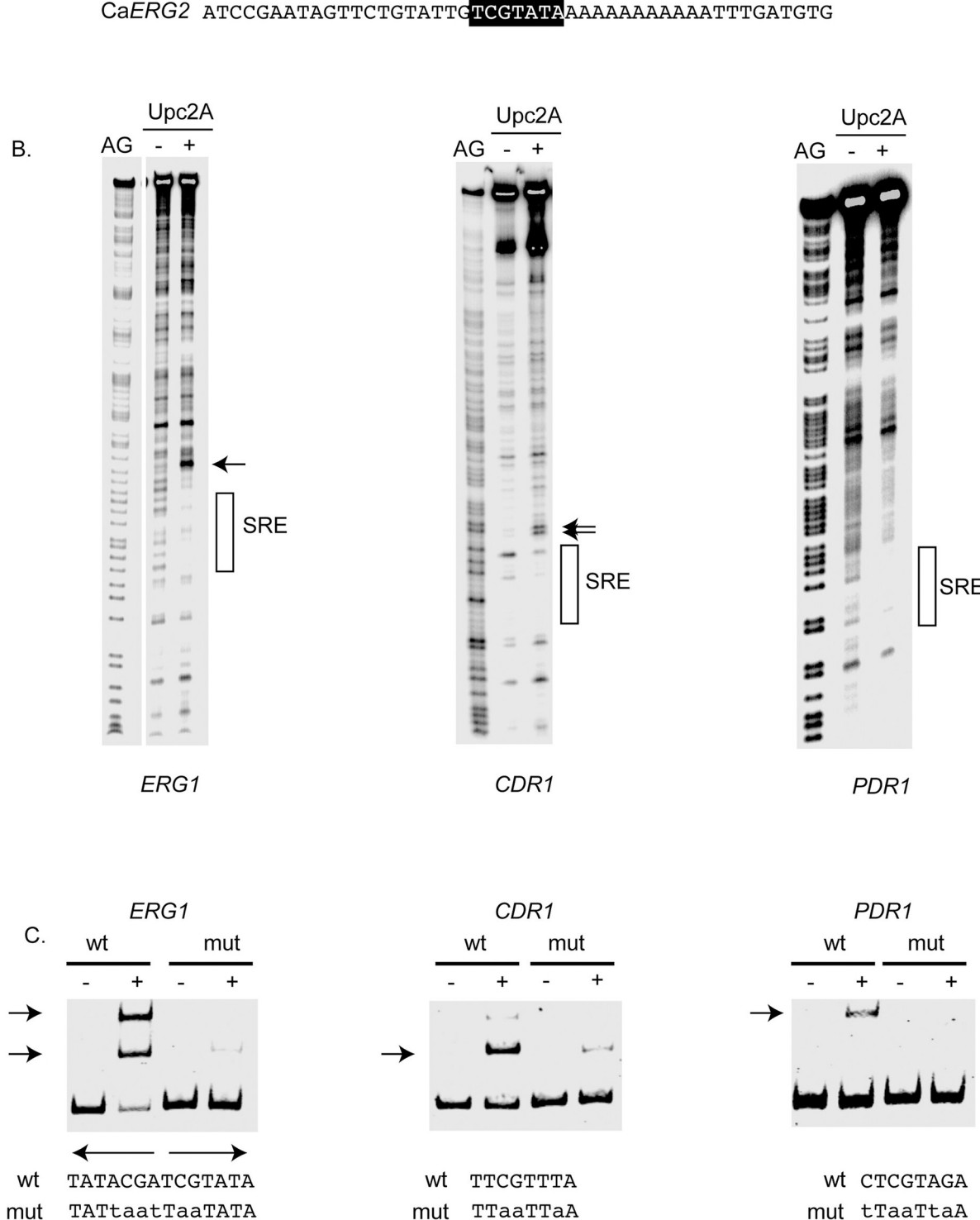

A.

| | |
|---|---|
| *ERG1* | CGACTCTATCACGTATACGATCGTATACGTGTGCTACCAACACCCAG |
| *CDR1* | TACCGGAACACGATTAGCCTTCGTTTACATGTTGGCAAACCCAGGAC |
| *PDR1* | TCTGTGCTTCATTTTCTACCTCGTAGATTAGGTTACGTTCAAATTTT |
| *FKS1* | CAGGGCTACTGCATTTTCTATCGTAACCGAACAAGTGAGAATTAGAA |
| Sc*ERG2* | GCAGAATCGAACCACGGCCCTCGTATAAGCCGCAAGGAAAACTACC |
| Ca*ERG2* | ATCCGAATAGTTCTGTATTGTCGTATAAAAAAAAAAAAATTTGATGTG |

B.

ERG1   CDR1   PDR1

C.

*ERG1*

| | wt | | mut | |
|---|---|---|---|---|
| | − | + | − | + |

wt   TATACGATCGTATA
mut   TATtaatTaaTATA

*CDR1*

| | wt | | mut | |
|---|---|---|---|---|
| | − | + | − | + |

wt   TTCGTTTA
mut   TTaaTTaA

*PDR1*

| | wt | | mut | |
|---|---|---|---|---|
| | − | + | − | + |

wt   CTCGTAGA
mut   tTaaTtaA

**Fig 4. DNA-binding of Upc2A to SREs in target genes.** A. Locations of SREs from several different promoters are shown. *ERG1*, *CDR1*, *PDR1* and *FKS1* all correspond to the *C. glabrata* locus while Sc*ERG2* is from *S. cerevisiae* and Ca*ERG2* is from *C. albicans*. Note that *ERG1* contains an everted SRE repeat indicated by the arrows. The single SRE is shown in black and white. The extent of DNA protected from cleavage by the DNase I mapping experiment (see below) is shown in gray. B. DNase I protection of the indicated *C. glabrata* promoters is shown. The position of each SRE is indicated by the bar at the righthand side and DNase I hypersensitive sites are noted by the arrows. AG refers to the purine-specific reaction of Maxam-Gilbert chemical sequencing and is carried out on the same radioactive DNA fragment used in the DNase I reaction. Recombinant Upc2A was added (+) to the DNA probe or omitted (-) as indicated. C. Electrophoretic mobility shift assay (EMSA) analysis of Upc2A binding to wild-type and mutant SREs. Biotinylated probes were prepared from the indicated C. glabrata promoter regions containing either wild-type (wt) or mutant (mut) versions of each SRE. Sequences of these different SREs are shown at the bottom of the panel with the altered residues in lower case. The SRE repeats in the *ERG1* promoter are shown by the divergent arrows at the top of the sequence. Position of the shifted protein:DNA complexes are shown by the arrows at the lefthand side of each image. The presence or absence of Upc2A protein is indicated by the (+) or (-), respectively.

transformed into wild-type cells, then grown to mid-log phase and challenged with or without fluconazole. *C. glabrata* promoter-dependent β-galactosidase activity was then determined.

Introduction of the mSRE into *ERG1*, *CDR1* or *PDR1* promoters led to a reduction in the level of fluconazole-induced β-galactosidase activity produced by each respective fusion gene (Fig 5A). While some degree of fluconazole inducibility was retained in each mSRE-containing promoter, these data indicate that each SRE identified above is required for normal drug induced promoter activation.

To examine the effect of the loss of the SRE from the wild-type *CDR1* and *PDR1* genes, the mSRE mutations were introduced into otherwise wild-type versions of these two genes. Isogenic wild-type and mSRE versions of the *PDR1* locus were prepared by recombination into the normal chromosomal location of this gene in a strain containing a null allele of *CDR1*. Low-copy-number plasmids containing *CDR1* were constructed that varied only by the form of the SRE that was contained in the promoter region. These two different forms of *CDR1* were introduced into the wild-type and mSRE *PDR1 cdr1Δ* strain and transformants grown in the presence or absence of fluconazole. Whole cell protein extracts were prepared and examined for expression of proteins of interest using appropriate polyclonal antisera (Fig 5B).

Loss of the SRE from the *CDR1* promoter caused a significant drop in expression when cells were treated with fluconazole that was enhanced when combined with the mSRE version of the *PDR1* gene. Similar reductions in Cdr1 levels were seen in the absence of fluconazole, again with removal of the SRE from both *CDR1* and *PDR1* causing the largest reduction in Cdr1 levels. Expression of Pdr1 was not affected when the *CDR1* SRE was removed but fluconazole induction of Pdr1 was reduced when the SRE was removed from the *PDR1* promoter. Expression of Erg11 was unaffected in these backgrounds as these alterations were restricted to the *CDR1* and *PDR1* promoters.

These strains were also evaluated for their drug resistance phenotype using a serial dilution assay on fluconazole-containing media (Fig 5C). The major reduction in fluconazole resistance was caused by the presence of the mSRE-containing form of the *CDR1* gene. This was modestly enhanced by the simultaneous loss of the SRE from the *PDR1* gene. Together, these data demonstrate that the SREs present in *CDR1* and *PDR1* are required for normal expression of these genes and for full fluconazole resistance.

## Role for Upc2A in caspofungin resistance

The ChIP-seq data predicted a potential SRE upstream of the *FKS1* and *FKS2* genes (S1 and S2 Tables). To determine if these putative SREs had detectable roles in expression of the caspofungin resistance phenotype, we tested the ability of an isogenic set of strains varying in their *UPC2A* allele for the response to several different cell wall stress agents. Isogenic wild-type, *upc2AΔ* or epitope-tagged wild-type or G898D *UPC2A*-containing strains were tested for

**A.**

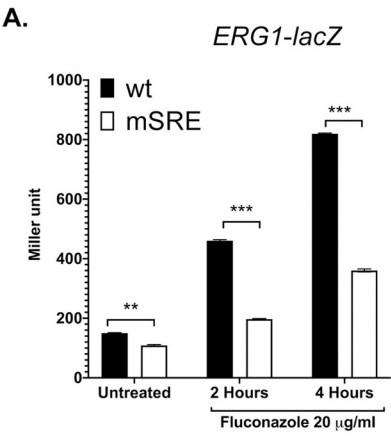
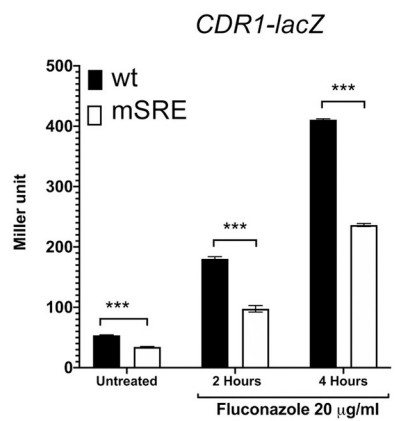
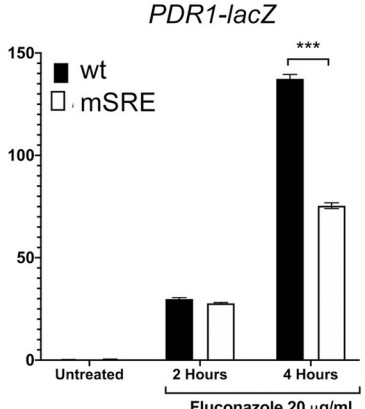

**B.**

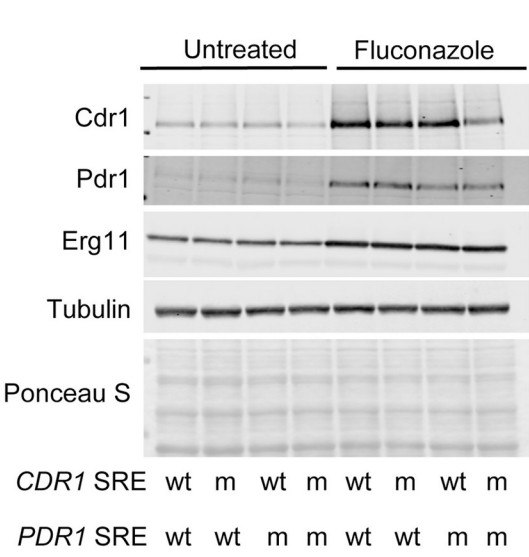
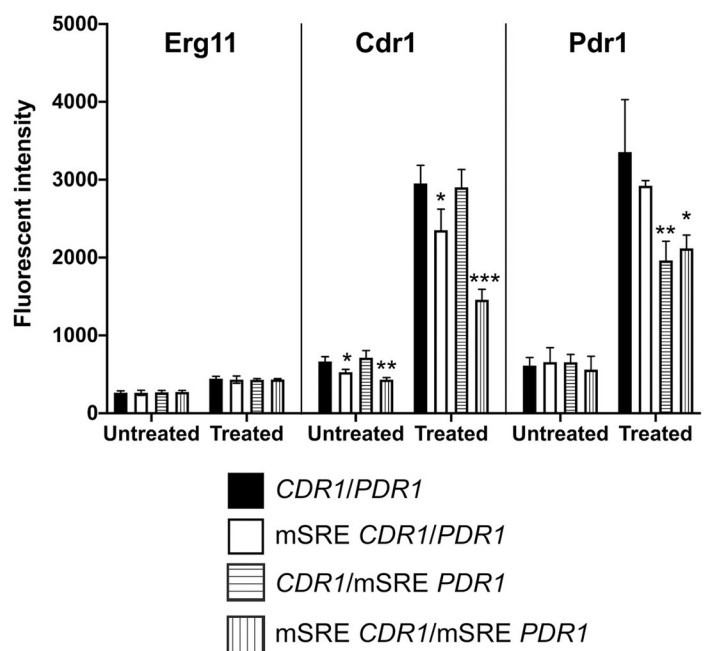

**C.**

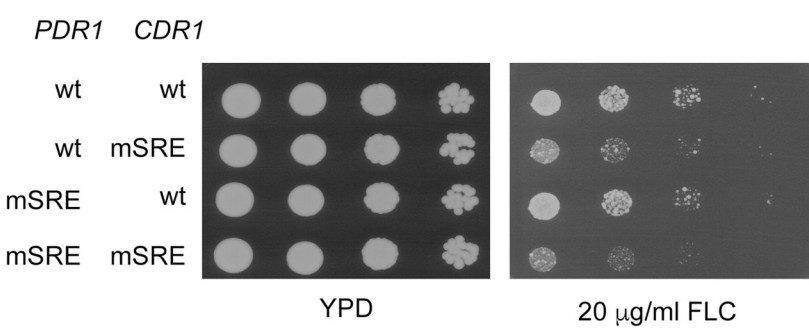

**Fig 5. Phenotypes of SRE mutations.** A. Normal expression of *lacZ* gene fusions requires the presence of intact SREs in Upc2A target promoters. Low-copy-number plasmids containing translational fusions between *ERG1*, *CDR1* and *PDR1* 5' regulatory region and *E. coli lacZ* were generated containing either the wild-type version of each promoter or the same fragment with the SRE mutant (mSRE) shown to reduce in vitro binding in Fig 4. These plasmids were introduced into wild-type cells, grown in the absence or presence of fluconazole (20 μg/ml) and then β-galactosidase activity determined. B. Western blot analysis of *CDR1* and *PDR1* expression upon loss of the wild-type SRE. All alleles of *PDR1* were integrated into the chromosome while all alleles of *CDR1* were carried on a low-copy-number plasmid. The presence of either the wild-type (wt) or mutant (m) SRE at each gene is indicated at the bottom of the panel. Each strain was grown in the presence or absence of fluconazole and levels of proteins of interest determined using western blotting with appropriate antibodies as described above. Erg11 was detected using an anti-peptide antiserum. The right hand panel shows the quantitation as described in Fig 1. C. The strains described above were tested by serial dilution for their growth on rich medium (YPD) or the same medium containing fluconazole (FLC).

resistance to caspofungin, caffeine or high pH using a serial dilution assay. Caffeine and high pH are cell wall stresses and reflect general cell wall dysfunction [34].

Loss of *UPC2A* caused hypersensitivity to all these agents (Fig 6A) while both epitope-tagged alleles behaved like the wild-type strain. The finding of a caspofungin susceptible phenotype prompted us to examine expression of the three *FKS* genes in *C. glabrata* to determine if any of these showed a response to the G898D allele of *UPC2A*. None of these genes were altered in the presence of this gain-of-function form of *UPC2A* while both *ERG1* and *AUS1* were elevated (Fig 6B), confirming the functionality of this hypermorphic form of Upc2A.

To explain the observed caspofungin hypersensitivity of the *upc2AΔ* strain, levels of *FKS1* and *FKS2* mRNAs were measured using RT-qPCR in the presence or absence of caspofungin. Loss of *UPC2A* reduced basal expression of *FKS1* by 50% and had a modest effect on *FKS2* (30% reduction) (Fig 6C). The addition of caspofungin strongly induced *FKS2* expression as expected [35] with this induction unaffected by the absence of Upc2A.

Since a defect was seen for *FKS1* expression, we prepared a DNA probe containing the putative SRE in this promoter for use in an EMSA to determine if recombinant Upc2A was able to recognize this element in vitro (Fig 6D). A mutant form of this SRE was also tested in this EMSA. The wild-type *FKS1* probe was strongly reduced in mobility when incubated with Upc2A while the mSRE-containing probe exhibited a band of reduced intensity upon loss of this sequence element.

To determine if the SRE was required for normal expression of *FKS1*, a *lacZ* translational fusion gene was prepared in which the *FKS1* regulatory region determined expression of β-galactosidase. Both the wild-type and mSRE-containing *FKS1* promoters were used and introduced on a low-copy-number plasmid into wild-type *C. glabrata* cells. *FKS1*-dependent β-galactosidase activities were then determined in the presence or absence of caspofungin induction.

Loss of the SRE from the *FKS1* promoter caused a significant reduction in *FKS1*-dependent expression of *lacZ* in the absence of caspofungin (Fig 6E). These data provide evidence that Upc2A-mediated gene activation is required for normal expression of *FKS1* and wild-type caspofungin resistance.

## Discussion

These data provide important new appreciation for the expansive role of Upc2A in control of gene expression. Extensive previous work on Upc2A homologues in both *S. cerevisiae* and *C. albicans* was generally done prior to the availability of modern genomic approaches like ChIP- and RNA-seq (reviewed in [36]). Detailed analyses demonstrated the crucial role of these Upc2A-like factors in regulation of *ERG* gene biosynthesis [30,37,38] but little was known about the full range of their target genes. A ChIP-chip experiment was carried out on *C. albicans* Upc2 and this factor was found to associate with the *CDR1* gene in this species [39]. To the best of our knowledge, there has been no follow-up linking *C. albicans* Upc2 with the Tac1

A.

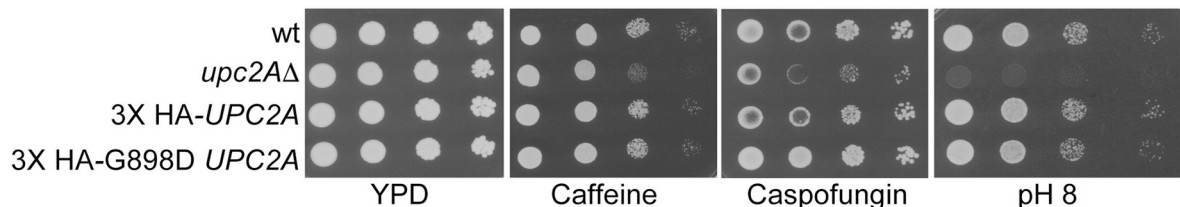

B.

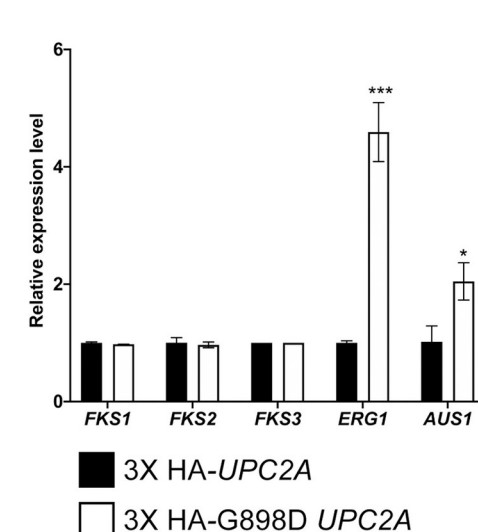

3X HA-*UPC2A*
3X HA-G898D *UPC2A*

C.

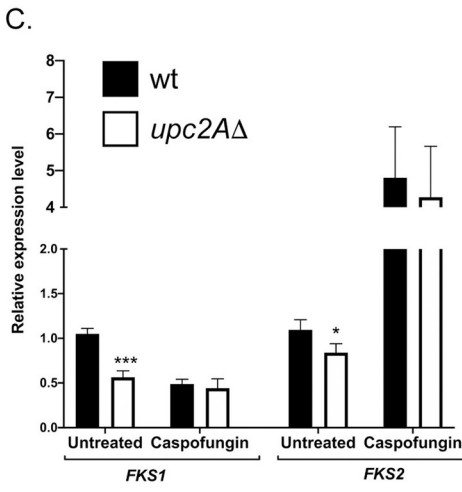

D.

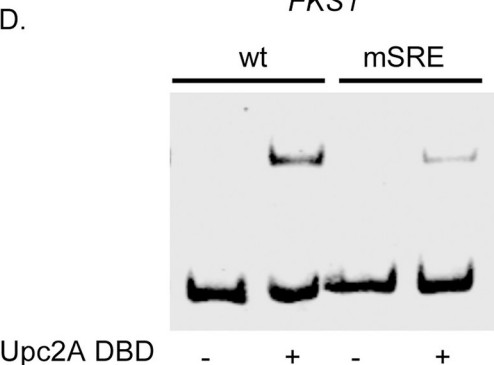

*FKS1*

wt    mSRE

Upc2A DBD    -    +    -    +

E.

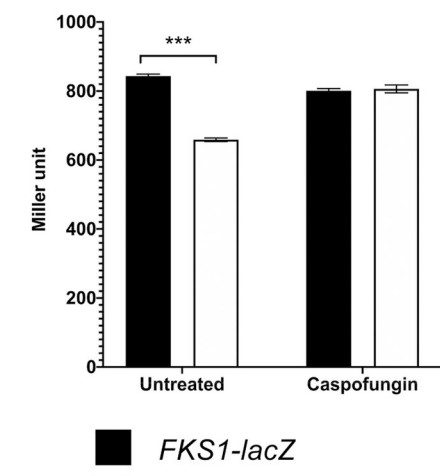

*FKS1-lacZ*
mSRE *FKS1-lacZ*

**Fig 6. Upc2A transcriptionally regulates *FKS1* expression.** A. An isogenic series of strains with the listed *UPC2A* genotypes was tested for resistance to the indicated stress agents that affect the cell wall. Strains were grown to mid-log phase and then serially diluted across each plate. Caspofungin was included at 100 ng/ml. B. Strains containing the different alleles of *UPC2A* were grown and analyzed for levels of the indicated RNAs using RT-qPCR. C. Isogenic wild-type and *upc2AΔ* strains were grown in the presence or absence of caspofungin and levels of *FKS1* and *FKS2* RNA assayed as above. D. A probe from the FKS1 promoter containing the putative SRE shown in Fig 4A was used in a EMSA experiment. A version of this probe lacking the SRE (mSRE) was also compared for its behavior in this assay. Upc2A was either omitted (-) or added (+) to the reaction prior to electrophoresis. E. An *FKS1-lacZ* fusion gene containing either the wild-type or the mSRE version of the promoter was introduced into wild-type cells. Levels of *FKS1*-dependent β-galactosidase were determined in the presence or absence of caspofungin.

transcription factor (key regulator of *CDR1* transcription) or *S. cerevisiae* Upc2 with either Sc*PDR1* or Sc*PDR3*. This suggests the possibility that this *C. glabrata* connection between Upc2A and Pdr1 is a unique feature of this yeast and could help explain the high level of intrinsic azole resistance seen in this organism.

The large number of Upc2A target genes illustrates the breadth of processes that are transcriptionally influenced by this factor. Clearly, the *ERG* genes are an important set of genetic targets but these are a small fraction of the whole. Upc2A appears to be coordinating a broad group of genes including a large number of plasma membrane-localized proteins (See S1 and S3 Tables). Coupled with its control of ergosterol in this membrane, Upc2A appears to be a central determinant of the composition of this membrane compartment in *C. glabrata*. The regulation of plasma membrane constituents is of obvious importance in modulating the ability of substances to cross this barrier between the external and internal environments.

Comparing the members of the target gene sets defined by Upc2A and Pdr1 suggests a hierarchical relationship between these two transcription factors. Here we establish that a binding site for Upc2A lies upstream of *PDR1* and is required for normal activation of *PDR1* expression (Fig 5) as well as many of the other genes controlled directly by Pdr1. We suggest Upc2A provides overarching control of both the Pdr1 regulon but also a variety of other genes that are not under Pdr1 control, serving to link these different classes of genes through this common transcriptional regulation.

While the full range of Upc2A target genes illustrate the global importance of this transcription factor, the *ERG* genes are especially sensitive to the level of activity of this regulator. The G898D *UPC2A* allele has a surprisingly limited effect on gene expression as this allele was seen to trigger significant transcriptional changes almost exclusively in genes associated with ergosterol biosynthesis (Table 2). It is possible that the transcriptional impact of the G898D Upc2A could be expanded if cells were treated with fluconazole as ergosterol limitation might be able to impact expression by regulatory inputs beyond Upc2A. Experiments to test this possibility are underway.

Construction and analysis of G898D Upc2A demonstrated that there is no particular prohibition on hyperactive alleles of *UPC2A* existing in this pathogenic yeast. The *S. cerevisiae* *UPC2-1* allele, from which we derived the G898D *UPC2A*, was originally isolated on the basis of permitting aerobic sterol uptake [19]. Strikingly, no other *S. cerevisiae* *UPC2* hypermorphic alleles are known. *UPC2* mutations in *C. albicans* have been found in multiple clinical strains and appear to be much more commonly isolated [23,24]. An interesting feature of the majority of *C. albicans* *UPC2* GOF alleles is these all cluster with a region of the protein between residues 642 and 648 [16]. This region shows strong sequence conservation with the 893–898 region of *C. glabrata* Upc2A. The conserved location of these hypermorphic alleles suggests the possibility that a common function is being disrupted in both organisms. The limited spectrum of genes that is induced by G898D Upc2A may reflect more complex requirements at these promoters for their activation beyond Upc2A occupancy.

Based on detailed structural and subcellular localization studies on *S. cerevisiae* Upc2 [22], we propose that *C. glabrata* Upc2A accumulates in the nucleus upon ergosterol limitation.

Our data support this assertion in two ways. First, a large increase in ChIP-seq peaks is seen for Upc2A when azole-treated cells are compared to untreated cells. Second, the G898D Upc2A mutant protein shows constitutively high number of ChIP-seq peaks that is not significantly altered by azole challenge. These data are consistent with a model in which Upc2A nuclear accumulation is enhanced upon ergosterol limitation and this regulation requires the function of the region containing G898 in the C-terminus of this factor. The molecular basis underlying the transcriptional response to G898D is still only partially understood and will require more detailed analysis to determine how this mutant allele of *UPC2A* leads to gene activation.

The finding of the interaction between Upc2A and the *FKS1* gene provides an interesting connection between azole resistance, well-known to be impacted by Upc2A [40], and echinocandin resistance. These two antifungal drugs have been considered to be defined by genetically separable pathways but here we provide evidence that Upc2A may provide a link between them. Intervention in Upc2A-mediated transcriptional activation may be able to cause reductions in resistance to both azole drugs and the echinocandins.

Finally, our data also illuminate the complexity and interrelationship of expression of genes involved in ergosterol biosynthesis with plasma membrane proteins and even the cell wall. *ERG* gene regulation is an important task for Upc2A but this factor clearly impacts transcription of a broad range of genes affecting multiple aspects of the plasma membrane. Additionally, loss of Upc2A has clear phenotypes but is certainly not the only regulator of *ERG* gene expression and fluconazole induction in *C. glabrata*. Loss of *UPC2A* leads to a profound increase in fluconazole susceptibility, even in the presence of a GOF form of *PDR1* [17]. However, only ~50 genes were both bound by Upc2A and dependent on Upc2A for fluconazole induction while 880 were induced in the presence of fluconazole independent of Upc2A (Fig 3A). This provides an illustration of the overlapping modes of regulation controlling gene expression in response to ergosterol limitation. The importance of ergosterol production and its synchronization with biogenesis of membrane proteins in the plasma membrane is central to a fungal cell producing a normally functioning membrane that can allow growth. Understanding this regulatory circuitry will allow interventions to be developed that can restore and potentially even enhance azole susceptibility, allowing the use of this highly effective antifungal drug to be maintained.

## Materials and methods

### Strains and growth conditions

*C. glabrata* was routinely grown in rich YPD medium [1% yeast extract, 2% peptone, 2% glucose] or under amino acid-selective conditions in complete supplemental medium (CSM) (Difco yeast nitrogen extract without amino acids, amino acid powder from Sunrise Science Products, 2% glucose). All solid media contained 1.5% agar. Nourseothricin (Jena Bioscience, Jena, Germany) was supplemented to YPD media at 50 µg/ml to select for strains containing the pBV133 vector [26]) and its derivatives. All strains used in this study are listed in Table 1.

### Plasmid construction and promoter mutagenesis

All constructs used for homologous recombination into the chromosome were constructed in a pUC19 plasmid vector (New England Biolabs, Ipswich, MA). PCR was used to amplify DNA fragments and Gibson assembly cloning (New England Biolabs) employed to assemble fragments together. All isogenic deletion constructs were made by assembling the recyclable cassette from pBV65 [26] and fragments from the immediate upstream/ downstream regions of the target genes. Eviction of the recyclable cassette left a single copy of loxP in place of the

excised target gene coding region. Sequences of the repeated influenza hemagglutinin epitope tag (3X HA) was PCR amplified from BVGC3 background [26]. This tag element was inserted before the start codon of *UPC2A* and G898D *UPC2A* with an addition of repeated 3X glycine-alanine linker sequence located between the 3X HA tag and the gene coding sequence. The G898D mutation in *UPC2A* was made by Gibson assembly in which the overlapping primers contained the point mutation sequence.

Gene complementation constructs were made by Gibson assembling the fragments from the immediate upstream and downstream regions of the target genes [overlapping regions], coding region of the target genes, target gene terminators (about 250 base-pairs after the translation stop codon), and the recyclable cassette (located after the terminator). Eviction of the recyclable cassette in the complementation constructs left a single copy of loxP about 250 base-pairs downstream of the target gene stop codons. Complementation of *LEU2* was done by PCR amplifying the *LEU2* coding region and 500 base-pairs immediate upstream and downstream of the coding region from the CBS138 background. Linear DNA was then transformed into KKY2001 and the colonies were selected on CSM agar without Leucine.

All autonomous plasmids were derived from pBV133 [26] carrying nourseothricin marker. The *lacZ* gene encoding the *E. c*oli β-galactosidase gene was amplified from pSK80 [41]. *ERG1*, *CDR1*, *PDR1* promoter fragments were amplified from the KKY2001 background. The *CDR1* minimal promoter, which was fused to *lacZ*, contained the -1 to -1076 region (with the ATG of *CDR1* considered as +1). The full *CDR1* promoter, which was used in the complementing plasmid, contained the -1 to -1504 region. The *PDR1* promoter contained the -1 to -847 region in all constructs. *ERG1* promoter region consisted of the -1 to -916 region and the *FKS1* promoter contained the -1 to -1795 region. SRE mutations in the target gene promoters were done by modifying the SRE core sequence and 2 adjacent bases into a PacI restriction enzyme sequence with Gibson assembly in which the overlapping primers contained the PacI sequence. All constructs were verified by Sanger sequencing (University of Iowa Genomic Core)

## *C. glabrata* transformation

Cell transformations were performed using a lithium acetate method [42]. After being heat shocked, cells were either directly plated onto selective CSM agar plates (for auxotrophic complementation) or grown at 30˚C at 200 rpm overnight (for nourseothricin selection). Overnight cultures were then plated on YPD or CSM agar plates supplemented with 50 μg/ml of nourseothricin. Plates were incubated at 30˚C for 24 to 48 h before individual colonies were isolated and screened by PCR for correct insertion of the targeted construct.

## Expression and purification of Upc2A DNA binding domain

The DNA sequence corresponding to the first 150 amino acids of the N-terminus of Upc2A was amplified by PCR and cloned into pET28a+ vector [digested with NcoI and SacI] with Gibson cloning. Correct clones were sequenced verified and transformed into the BL21 DE3 *E. coli* expression strain (Thermo Fisher, Waltham, MA). Mid-log phase cells were induced with 1 mM IPTG (Fisher Scientific, Hampton, NH) for 4 hours at 30˚C. Collected cells were lysed using a French Press G-M high pressure disruptor (GlenMills, Clifton, NJ). The clarified lysate was subjected to Talon Metal Affinity column (Takara, Mountain View, CA) as per the manufacturer's protocol. Purified protein was dialyzed with dialysis buffer [20 mM Tris pH 8.0, 500 mM NaCl, 1 mM dithiothreitol and 0.5% Tween 20] for 24 hours and its concentration was quantified by Bradford assay (Bio-Rad, Des Plaines, IL).

## Quantification of transcript levels by RT-qPCR

Total RNA was extracted from cells by extraction using TRIzol (Invitrogen, Carlsbad, CA) and chloroform (Fisher Scientific, Hampton, NH) followed by purification with RNeasy minicolumns (Qiagen, Redwood City, CA). RNA was reverse-transcribed using an iScript cDNA synthesis kit (Bio-Rad, Des Plaines, IL). Assay of RNA via quantitative PCR [qPCR] was performed with iTaq universal SYBR green supermix (Bio-Rad). Target gene transcript levels were normalized to transcript levels of 18S rRNA during fluconazole challenge and β-tubulin mRNA in other conditions. Primer sequences were listed in S4 Table. Data reported are from two biological replicates with two technical replicates of each.

## Spot test assay

Cells were grown in YPD medium to mid-log-phase. Cultures were then 10-fold serially diluted and spotted onto YPD agar plates containing different concentrations [10 or 20 µg/ml] of fluconazole (LKT laboratories, St Paul, MN), caspofungin 100 ng/ml (Apexbio, Houston, TX), congo red 100 µg/ml (Sigma-Aldrich, St. Louis, MO), caffeine (Sigma-Aldrich). In some experiments, the YPD medium and agar was supplemented with 50 µg/ml nourseothricin to maintain plasmids derived from the pBV133 vector [26]. All agar plates were incubated at 30˚C for 24 to 48 h before imaging was performed. To adjust the pH level, 100 mM of MES (VWR, Radnor, PA) [pH 5.5], HEPES (RPI) [pH 7.0], and TAPS (Sigma-Aldrich) [pH 8.0] were added to the 2x YPD. Solutions were then filtered and mixed with autoclaved 3% agar to make YPD agar plates.

## Electrophoretic mobility shift assays (EMSA)

DNA probes were amplified by PCR with biotinylated primers (IDT, Coralville, IA) corresponding to the sequences listed in S2 Table. Fragments from the *ERG1* promoter -704 to -916, *CDR1* promoter -560 to -731, *PDR1* promoter -552 to -728, *FKS1* promoter -1489 to-1666, and *HO* promoter -787 to-957 regions were amplified. Reaction buffer [18 µl], containing 5 µg sheared salmon sperm DNA (Thermo Fisher Scientific, Waltham, MA), 5% Glycerol, 0.01% NP40, 0.1% bovine serum albumin (Thermo Fisher Scientific), and 2 µl of 10x binding buffer [100 mM Tris pH 7.5, 400 mM NaCl, 10 mM DTT and 100 µM ZnSO$_4$], was incubated with different concentrations of Upc2A-6X His or 1X binding buffer for 10 minutes at room temperature. Biotinylated probes [20 fmol] were added in a final reaction volume of 20 µl and incubated for additional 20 min at room temperature. Samples were immediately subjected to electrophoresis on 5% polyacrylamide Tris/Borate/EDTA [TBE] gel in 0.5x TBE running buffer at 4˚C. Subsequently, samples were transferred into a nylon membrane (GE, Chicago, IL) in 0.5X TBE buffer at 4˚C. Samples were then crosslinked on nylon membrane under UV light for 10 min. Membrane was blocked with Intercept blocking buffer (LI-COR Biosciences, Lincoln, NE) containing 1% SDS for 30 min before IRDye 680LT Streptavidin (LI-COR Biosciences) antibody was added at 1:20000 final dilution. After 35 min of incubation, the membrane was washed three times with phosphate buffer saline [PBS] containing 0.1% tween (RPI). Imaging was performed with Odyssey CLx Imaging System (LI-COR Biosciences) and analyzed by Image Studio Lite Software (LI-COR Biosciences).

## Chromatin immunoprecipitation-Next Generation Sequencing

Overnight cultures were inoculated at 0.1 OD/ml in fresh YPD and allowed to grow to 0.4–0.5 OD/ml. Cells were treated with Fluconazole 20 µg/ml or ethanol for 2 hours. Cells were fixed with 1% formaldehyde (Sigma-Aldrich) for 15 min at room temperature with mild shaking

[120–150 rpm]. The fixing reaction was stopped with 250 mM glycine (RPI) for 15 min at room temperature with mild shaking [120–150 rpm]. Cells were centrifuged and washed once with PBS. The cell pellet was resuspended in lysis buffer [50mM Hepes pH 7.5, 140mM NaCl, 1mM EDTA, 1% Triton X-100, 0.1% Sodium deoxycholate, 1mM PMSF, 1x protease inhibitor cocktail (Roche Applied Science, Penzberg, Germany). Cells were then lysed with 1 ml glass beads (Scientific Industries Inc, Bohemia, NY) at 4°C for 10 min. Both cell lysate and debris were collected and subsequently transferred to an AFA fiber pre-slit snap-cap [6 x 15mm] microtube (Covaris, Woburn, MA) for additional cellular lysis and DNA shearing.

Genomic DNA was sheared with E220 focused-ultrasonicator (Covaris) [peak incident power: 75 W, duty factor: 10%, cycles per burst: 200, treatment time: 16 min, temperature: 10°C max, sample volume: 130 μl.] The sheared sample was centrifuged and the clear lysate was collected. Upc2A was immunoprecipitated with Dyna beads-protein G magnetic beads (Invitrogen, Carlsbad, CA) and anti-HA antibody (Invitrogen) [1:50 dilution] overnight at 4°C. Beads were then washed twice with lysis buffer, once with lysis buffer+ 500mM NaCl, once with LiCl buffer [10 mM Tris pH 8, 250 mM LiCl, 0.5% P-40, 0.5% Sodium deoxycholate, and 1 mM EDTA], and once with Tris-EDTA buffer. Beads were resuspended in TE and treated with RNAse A (Thermo Fisher Scientific) at 37°C for 30 min. Beads were then washed once with Tris-EDTA buffer, resuspended in Tris-EDTA buffer with 1% SDS, and incubate at 65°C for at least 5 hours to reverse crosslink. Eluted DNA was subsequently purified with mini elite clean-up kit (Qiagen). Qubit fluorometric assay (Thermo Fisher Scientific) was used to analyze the yield quantity and Agilent Bioanalyzer (Agilent Technology, Santa Clara, CA) was used to determine the average sheared DNA size.

All ChIPed DNA libraries were generated with Accel-NGS plus DNA library kit (Swift Biosciences, Ann Arbor, MI) according to the manufacturer's instructions. Samples were sequenced at the University of Iowa Institute for Human Genetics Genomics Division using an Illumina NovaSeq 6000 instrument. Read quality was confirmed using FastQC (Babraham Bioinformatics). The reads from duplicate experiments were combined and mapped to the *C. glabrata* CBS138 genome using HISAT2 [43]. Paired reads with intervening fragments greater than 1000 bp were removed during the mapping. The total number of mapped reads were reduced by randomly selecting 2.5% from each bam file which were then sorted and indexed using Samtools. ChiP-seq peak calling was done with the callpeak function of MACS2 using a false discovery rate (q-value) cutoff of 0.001 and a maximum allowable gap between peaks of 100 bp [27]. Differential peak detection among the experimental conditions was done using the bdgdiff function of MACS2. The default likelihood ratio cutoff of 1000 was used. Output files from both callpeak and BDGdiff were annotated to identify candidate downstream genes using the ChIPpeakAnno R package [44]. All large datasets (both ChIP- and RNA-seq) are available as a GEO reference Series: GSE182516.

## RNA-sequencing

A single colony of each *C. glabrata* strain was used to inoculate 2 ml of YPD, which was grown overnight at 30° C in an environmental shaking incubator. Cell density was then adjusted to $OD_{600}$ = 0.1 in 10 ml YPD, and cultures were grown as before for 6 hrs (mid-log phase). For the fluconazole-treated strains, either fluconazole (50 μg/ml final concentration) or DMSO (diluent control) was added to the 10 ml culture and grown for 6 hrs. Cells were collected by centrifugation, supernatants discarded, and cell pellets stored at -80° C. RNA was isolated from cell pellets via a hot phenol method as described previously [45]. The quantity and purity of RNA were determined by spectrophotometer (NanoDrop Technologies, Inc., Wilmington, DE) and verified using a Bioanalyzer 2100 (Agilent Technologies, Santa Clara, CA). Library

preparation and RNA sequencing analysis were performed as previously described [46]. Transcript quantification of expression levels and analysis of differential expression were done using HISAT2 and Stringtie [47]. Differential expression was analyzed using DESeq2 [48].

## DNase I protection assay

DNA probes were generated by PCR. To generate a 5′ [γ-$^{32}$P] singly end-labelled probe one of the PCR primers was modified [5 Amino-MC6 (Integrated DNA technologies, Coralville,IA)] at the 5′ end to prevent phosphorylation by polynucleotide kinase. Probes were end-labelled [1 pmol] using 10 μCi of [γ-$^{32}$P]-ATP (PerkinElmer, Waltham, MA) and 10 U polynucleotide kinase (New England Biolabs, Beverly, MA) as instructed by the manufacturer. Unincorporated [γ-$^{32}$P]-ATP was removed using a nucleotide removal column (Qiagen). The binding reaction was done as described in the EMSA section, and the sample was digested with DNase I (NEB [1:20 dilution]) for 30 seconds at room temperature. DNase I foot-printing and DNA sequencing reactions were performed as previously described [49].

## β-galactosidase assay

Harvested cells were lysed with glass beads (Scientific Industries Inc) in breaking buffer [100 mM Tris pH8, 1 mM Dithiothreitol, and 20% Glycerol] at 4°C for 10 min. Lysate was collected and β-galactosidase reactions carried out in Z-buffer [60 mM $Na_2HPO_4$, 40 mM $NaH_2PO_4$, 10 mM KCl, 1 mM $MgSO_4$, 50 mM 2-Mercaptoethanol] with 650 μg /ml O-nitrophenyl-β-D-galactoside [ONPG]. Miller units were calculated based on the equation: [$OD_{420}$ x 1.7] / [0.0045 x total protein concentration x used extract volume x time]. The Bradford assay (Bio-Rad) was used to measure the total protein concentration in the lysate.

## Western immunoblot

Cells were lysed with lysis buffer [1.85 M NaOH, 7.5% 2-Mercaptoethanol]. Proteins were precipitated with 50% Trichloroacetic acid and resuspended in Urea buffer [40 mM Tris pH8, 8.0 M Urea, 5% SDS, 1% 2-Mercaptoethanol]. Cdr1, Pdr1, and Upc2A rabbit polyclonal antibodies were previously described [18,31]. Mouse anti-HA monoclonal antibody was purchased from Invitrogen. Secondary antibodies were purchased from LI-COR Biosciences. Imaging was performed with Odyssey CLx Imaging System (LI-COR Biosciences) and analyzed by Image Studio Lite Software (LI-COR Biosciences). Detected target band fluorescence intensity was normalized against tubulin fluorescence intensity and compiled from 2 biological replicate experiments and 2 technical replicates in each experiment, giving 4 replicates in total.

## Statistical analysis

Unpaired T-test was used to compare between isogenic deletion mutant and wildtype strains. Paired T-test was used to compare between the drug treated and non-treated conditions. *, **, *** were designated for $P \leqq 0.05, 0.01, 0.001$ respectively.

## Supporting information

**S1 Table. GO term enrichment in different classes of Upc2A-bound genes.** The 3 main categories of genes enriched from Fig 2A were analyzed for GO term enrichment using FungiFun < https://elbe.hki-jena.de/fungifun/fungifun.php > and default settings. A. GO terms enriched in genes bound only in wild-type cells challenged with fluconazole. B. GO terms from genes bound by Upc2A under all conditions. C. GO terms from genes bound by Upc2A under

induced conditions.
(XLSX)

**S2 Table. GO terms for genes bound by both Pdr1 and Upc2A.** Genes that were bound by Upc2A in the presence of fluconazole and by Pdr1 in cells lacking their mitochondrial genome ($\rho^0$) were analyzed by FungiFun as above.
(XLSX)

**S3 Table. Expression of genes bound by Upc2A in the presence of fluconazole.** A list of genes enriched in wild-type Upc2A-bound genes in the presence of fluconazole versus the absence of this drug is populated with RNA-seq data for each gene under 3 different conditions. The log2 values of the ratios of expression detected for fluconazole-treated cells over the untreated control (Flu/con wt), fluconazole-treated upc2A null cells over untreated upc2A null cells (Flu/Con upc2A) and wild-type over upc2A null cells (con wt/con upc2A).
(XLSX)

**S4 Table. Primers used in this study.**
(XLSX)

## Acknowledgments

We thank Dr. Damian Krysan for helpful discussions.

## Author Contributions

**Conceptualization:** Bao Gia Vu, P. David Rogers, W. Scott Moye-Rowley.

**Data curation:** Bao Gia Vu, Mark A. Stamnes, W. Scott Moye-Rowley.

**Formal analysis:** Bao Gia Vu, Mark A. Stamnes, Yu Li, P. David Rogers, W. Scott Moye-Rowley.

**Funding acquisition:** P. David Rogers, W. Scott Moye-Rowley.

**Investigation:** Bao Gia Vu, Mark A. Stamnes, Yu Li, P. David Rogers.

**Methodology:** Bao Gia Vu, Mark A. Stamnes, Yu Li, P. David Rogers.

**Project administration:** P. David Rogers, W. Scott Moye-Rowley.

**Supervision:** P. David Rogers, W. Scott Moye-Rowley.

**Writing – original draft:** Bao Gia Vu, Mark A. Stamnes, P. David Rogers, W. Scott Moye-Rowley.

**Writing – review & editing:** Bao Gia Vu, Mark A. Stamnes, P. David Rogers, W. Scott Moye-Rowley.

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
