## [Decision Letter · Decision Letter 0]

30 May 2021

Dear Scott,

Thank you very much for submitting your Research Article entitled 'Azole resistance is mediated by integration of sterol gene regulation and membrane transporter production by the zinc cluster-containing transcription factor Upc2A in Candida glabrata' to PLOS Genetics.

The manuscript was fully evaluated at the editorial level and by three independent peer reviewers who are all experts in the field. The reviewers appreciated the attention to an important problem, but raised some substantial concerns about the current manuscript. Based on the reviews, we will not be able to accept this version of the manuscript, but we would be willing to review a much-revised version. We cannot, of course, promise publication at that time.  In particular, the reviewers felt that the presentation would benefit from significant editing to focus on the most salient findings and key associated information.  Reviewer 3 also raised a key experimental question (point 5) that addresses comparison of wild type and drug resistant mutant at the same drug concentration given that the strains grow very differently under these conditions, and this reviewer suggested ways this could be addressed experimentally.

If you decide to revise the manuscript for further consideration at PLOS Genetics, please aim to resubmit within the next 60 days, unless it will take extra time to address the concerns of the reviewers, in which case we would appreciate an expected resubmission date by email to plosgenetics@plos.org.

[LINK]

We are sorry that we cannot be more positive about your manuscript at this stage. Please do not hesitate to contact us if you have any concerns or questions.

Yours sincerely,

Joseph Heitman, MD, PhD

Associate Editor

PLOS Genetics

Gregory P. Copenhaver

Editor-in-Chief

PLOS Genetics

Reviewer's Responses to Questions

**Comments to the Authors:**

Reviewer #1: This manuscript contains the equivalent work of about 2 papers. Unfortunately it reads like about 4 papers.

The most significant finding in my opinion is that Upc2A controls expression of targets of both azoles AND echinocandins in C. glabrata. This finding is sensational because this feature of the Upc2A network may contribute to the high intrinsic resistance of Cg to both classes of drugs. This finding and the extensive analysis leading up to it make the study well worth publishing in PLOS Genetics in my opinion. (The data and logic seem very solid.) A second major finding is that the Upc2A and Pdr1 networks overlap extensively. This finding augments the value of the manuscript and is directly relevant to azole resistance. However, I recommend that the manuscript be rewritten almost entirely to focus on the key findings. I also recommend the elimination of one entire section and a lot of text.

The major presentation problems that I see are:

1. Excruciatingly unnecessary detail. A great example is lines 186 to 196, in which ChIP-seq is described. Honestly, anyone who reads this paper will know what ChIP-seq is. Another example is 138-148, in which making an epitope-tagged mutant protein is explained at the level required for college sophomores. In the Introduction, lines 107-122 are an almost verbatim repetition of the Abstract. There are many other examples. Please economize.

2. Data and bioinformatic tsunami. Lines 184-349 present the interpretation of numerous ChIP-seq experiments for both Upc2A and Pdr1 and, about midway through, some RNA-seq analysis as well. There are no subheadings. It is very tough to absorb. Please break it into digestible units. (One thought - can't you start with WT Upc2A and then in a subsequent section, present results with the GOF version? Does it all have to be interdigitated?)

3. Weakly connected section. The section on anaerobic growth (lines 423-470) is a nice body of work but it is only remotely connected to the rest of the study. My best advice is to get rid of this entire section. It would make a nice short paper all on its own in an admittedly lesser journal.

I have a more minor suggestion. The title in my opinion is too verbose and specialized. How about: Global role of Candida glabrata Upc2A in expression of drug target genes

Control of both azole and echinocandin drug target genes by Candida glabrata Upc2A

Candida glabrata Upc2A governs instrinsic azole and echinocandin drug resistance

There's a few different versions for your consideration.

I have one scientific question. Why in your opinion don't Upc2A GOF mutations arise during natural infection?

Reviewer #2: The paper aims to characterize the network of genes bound and regulated by the transcription factor Upc2A in C. glabrata by applying genome wide ChIP-seq and transcriptomic approaches. The authors extensively categorize the gene subsets that are bound and/or regulated by Upc2A as well as establish a link between these gens and the transcriptional network of Pdr1. They further characterize the importance of SRE in the regulation of several genes involved in azole and caspofungin response. The findings in this work will provide a solid resource for the future study of C. glabrata genetics and drug resistance.

Major suggestions

1. Several interesting observations are made with the ChIP-Seq. There are a large number of promoters (309) that are always bound by Upc2. There are also a large number of promoters that are bound in wild type upon fluconazole treatment and in the gain-of-function mutant (85). Yet this was a small fraction of the total number of promoters bound by Upc2 under FLC treatment. Was this small overlap expected? I would have thought for a true gain-of-function mutation, this overlap would be more substantial. Is there differences in the SRE promoter sequences under these different conditions?

2. For the RNA-seq data, the authors find 53 transcripts that require Upc2A for expression and that are bound by the regulator. These represent the direct targets. These should be more clearly described and highlighted in the text. Furthermore, I found it surprising that only 11 genes were significantly upregulated in the Upc2A gain-of-function (GOF) mutant. How do these compare with the 53 transcripts? Overall, my interpretation of this analysis is that Upc2A predominantly governs genes involved in the biosynthesis of ergosterol. The conclusion the Upc2A regulates other diverse sets of genes is likely largely indirect.

3. Ergosterol biosynthesis genes (ie ERG11) were shown to be exclusively bound by Upc2A (Fig. 2D) as opposed to being bound by both Upc2A and Pdr1 (ie CDR1). Therefore, it was puzzling in Fig. 5 when the authors introduced a mutant SRE sequence into the promoter regions of CDR1 and PDR1 but not ERG11. I would like to see the impact of mutating the SRE on the expression of a gene that is not also regulated by another key transcriptional regulator. How does mutating the ERG11 SRE impact fluconazole resistance? If there was a reason this was only performed in with ERG1 and not ERG11, this should be explained in text.

4. Similarly, for assessing the role of Upc2A in anaerobic growth, I would like to see how mutating the ERG11 SRE impacts the induction of ERG11 in aerobic versus anaerobic conditions.

5. From what I can tell, the only experiment that specifies number of replicates performed in the analysis is for the western blots. Please specify how many times the other experiments were performed. (Be specific regarding biological or technical analyses). This is especially important for qRT-PCR for which some of the “significant” changes are minimal.

6. Overall the authors should revise the manuscript to ensure language and background information is as clear as possible. For example, line 143-146 should specify that it replaced the native UPC2A and line 191 should explain the p0 cells are C. glabrata without mitochondria. Furthermore, there were several typos such as line 202, take out “be”; line 450, change “early” to “earlier”; and line 549, add “where” in between “from we”.

Minor suggestions

1. For Figure 1: Include alignment of S. cerevisiae and C. glabrata Upc2.

2. Line 77: I would recommend removing the word “successfully” when referring to azole therapy. Resistance does lead to clinical failure. Perhaps another word such as “extensively” would be better suited. Alternatively, you could highlight that its initial success has receded due to resistance.

3. It remains unclear why UPC2 gain-of-function mutations would have reduced PDR1 and CDR1 expression in presence of fluconazole. Can the authors speculate why this would be the case. Especially given that in Fig. 2D they demonstrate Upc2A occupancy at the CDR1 promoter in Upc2A wild type and GOF strains in the absence and presence of FLC.

4. The description of the genomic analyses (ChIP-Seq and RNA-seq) was confusing at times. Refining the text to provide a systematic and clear description of the findings would be beneficial.

5. For the competition experiment in Fig 6D, the authors should repeat the experiment such that the LEU2 marker is placed in the wild-type strain and the cdr1pdr1 double mutant is Leu-. This would confirm that it is not the LEU2 prototrophy that is impacting fitness. Additionally, it would be interesting to see if the results were differed when cdr1 and pdr1 mutants were assessed separately rather than just the double mutant. Especially given the finding that these transcriptional networks are tightly linked.

6. It is interesting that of the ergosterol biosynthesis genes significantly increased in the UPC2A GOF mutant, ERG4 (8, 9, 24, 26, 27)was the only one reported to be not directly bound by Upc2A. It would be interesting for the authors to address this point in the discussion.

7. What happens to FKS1 and FKS2 expression in the Upc2 GOF mutant? Especially given that a potential SRE was identified in the promoter region of these genes?

8. It would be interesting for the authors to speculate as to why the GOF UPC2 mutation only led to the upregulation of specific target genes.

9. In the results section the authors often chose to use language such as “strongly induced” or “reduced” (ex. line 489-490). The impact of these statements would be enhanced if they included quantitative metrics, such as “a 1.5-fold increase”.

10. The finding that PDR1 is regulated by Upc2A and contains a sensitive SRE is very interesting. Given that 64 genes were found to be transcriptionally fluconazole induced in a Upc2A-dependent manner but not directly bound by Upc2A, it would be interesting if the authors could address how many of these genes are regulated by PDR1.

Reviewer #3: Vu et al. describe genes that are regulated by the transcription factor UPC2 in the pathogenic yeast, Candida glabrata. They demonstrate that the transcription factor not only regulates genes in the ergosterol pathway but also regulate the transcription factor that controls efflux pump genes. They show that the UPC2 transcription factor is a “master” regulator of many pathways, and is involved in transcription associated with anaerobic growth and in Caspofungin susceptibility.

The data is clear and well presented, the interpretations are justified based on the data and the interpretations are for the most part valid. The paper is well written for the most part – issues are discussed below. The statistics are appropriate. The following comments could improve the significance of the publication

General Comments

1. The authors do not discuss UPC2B in Candida glabrata (Cg) or ECM22 in Saccharomyces cerevisiae (Sc). The presence of these paralogs is important in both species and needs to be addressed in this manuscript.

2. The manuscript suffers from clarity regarding which genes are being referred to. While there is only one UPC2A and it is from Cg, the manuscript would benefit greatly if it discussed CgUPC2A, CaUPC2, ScUPC2 etc. Or at least add the prefix to genes that are not Cg.

3. The authors use YPD as a growth media for most experiments. YPD has yeast extract, which contains ergosterol. Therefore, the cells can take up ergosterol from the media rather than synthesizing it de novo. Therefore, the effect of fluconazole on the cells will be muted by the use of YPD as a growth media for may experiments. This should be discussed in the paper.

4. The authors induce gene expression for 2 hours. Is there a justification for the two-hour induction? The use of YPD would blunt the effect of fluconazole, especially in a short induction period. This should be discussed in the paper.

5. In Figure 1A, the authors show that the WT is inhibited at 20 ug/ml of fluconazole, while the G898D mutant is resistant at that drug concentration. The authors then use 20 ug/ml as the concentration to study how the gene expression of WT and mutant differ in the presence of fluconazole. However, at this single drug concentration, the authors show in Fig 1A that the WT is highly stressed and unable to grow and the mutant is growing much better and its stress is limited. Therefore, the changes in gene expression at this one drug concentration can not be compared as the explanation is just that the WT and mutant cells were under different levels of drug stress. This is a significant problem throughout the paper. It could be solved by comparing WT at 8-fold above its MIC and the mutant at 8-fold above its MIC. That would be a fair comparison. I understand the amount of work it would require to redo all of the experiments in this paper. I would suggest a single RT-PCR experiment like Fig 1B in which the gene expression was monitored in WT and mutant at two different drug concentrations – one drug concentration 8-fold above the WT MIC and one drug concentration at 8-fold above the mutant MIC (maybe 4-fold above if the cells don’t grow well). Comparison of the RT-PCR under all these conditions would significantly address the issue (or indicate further study is needed)

Figures

6. In Figure 1 and subsequent figures, it does not mention what fluconazole concentration was used (with the exception of Figure 4). This should be in each figure legend.

7. Figure 2 –

a. 2A - the abbreviation WTF should not be used. No internet savvy reader can think of anything but what it stands for on the web. Maybe WT-F or WT-FLC

b. 2A - The labels for the 4 ovals should be more clearly attached to the oval – perhaps within the oval’s largest open space

c. 2B – The Sc sequences and the Cg sequences are presented in a different format. The sequences should be presented in a similar format. The left part of the Cg sequence appears to be an inverted repeat of the right part.

d. 2C – It would be useful to know where the 73 genes from 2C are located within the 2A Venn diagram.

e. 2D – The arrows at the bottom denoting the orientation of the gene are not at all clear. Larger arrows denoting the orientation of the gene should be added

8. Figure 3 – The Ven diagram is not well explained. The label for the green circle is not clear and needs to be changed. The text describing this figure needs to be much clearer.

9. Figure 4B right – As the text describes changes in Erg11 expression, as well as the other genes it might be best to present the Y axis as a log scale.

10. Figure 6A – The scale might better convey the results if the Y axis was presented as a log scale.

11. Figure 6D – The legend on the figure is difficult to read and it is organized in an unusual way. It might be better to place Aerobic WT and Aerobic pdr1/cdr1 on the left, one over the other and place the two anaerobic legends one on top of the other on the right, to parallel the actual location of the data. In addition, the shading is very difficult to discern and might be better in color or widening the columns in the graph or both.

12. Fig 7 – The concentrations of Caspofungin are not on the figure or figure legend.

Minor points

13. The standard ASM abbreviation for Fluconazole is FLC. That could be used throughout once defined.

14. In some experiments, FLC is dissolved in ethanol (line 691), in some it is dissolved in DMSO (line 775/6). This is problematic. It should be consistent.

15. Add reference to your figures in the text each time you describe relevant data from a figure.

16. While the agar dilution dots are adequate, the paper would be improved if it actually determined the MIC using a microbroth or macrobroth protocol.

17. Lines 77-78: Doesn’t make sense to say its success has led to development of resistance. Development of resistance would be a failure of therapy. Clarify by adding more info. Such as ‘Its abundant use/Its overuse has led to”

18. Lines 79-82: Add more clarification about what the epidemiology has changed FROM. From albicans infections to glabrata? Need to specify

19. Lines 84: What does high intrinsic resistance to FLC mean? High rates of intrinsic resistance in this species? High MIC?

20. Lines 91-93: “The range of genes… is much wider” is too vague. Use more specific and direct language.

21. Line 150- the resulting STRAIN exhibits elevated resistance- not the factor or the gene.

22. Lines 165-167 – The ERG11 gene may already be maximally expressed.

23. Line 168: ‘was enhanced’ I would consider the expression levels with the wild type transcription factor as the normal levels, and the levels with the mutant form the altered condition. So, the wording would be more accurate to say the GOF mutation inhibited/repressed/decreased/diminished or was less intense than the WT expression levels.

24. Line 168-169: I think more detailed description is important here. How much were the pdr1 and cdr1 genes induced by WT upc2A with fluconazole treatment? You specifically detailed in the next paragraph that upc2 was two-fold elevated. So, add in the details about pdr1, cdr1 and erg11.

25. Is this induction pattern with the wt CgUpc2a consistent with C albicans or Sc upc2 induction of CDR1 and pdr1 with fluconazole?

26. Is the lack of induction by fluconazole in the mutant GOF upc2a consistent with the mutant upc2 in albicans/sc?

27. Line 174: There is no Figure 1D

28. Lines 175-182: Should add ERG11 to figure 1c/d

29. Line 184: might be more accurate to say ‘the HA-tag did not appear to affect/disrupt function’

30. Line 191: What are the p0 cells? Need an introduction to them.

31. Lines193-194: What does ‘fixed chromatin prepared’ mean? Fix chromatin was prepared?

32. Line 195: antibodies

33. Line 196: change ‘by use of the’ to ‘using the’

34. Line 201: Instead of saying the wild-type factor, could say for the wild type CgUpc2A

35. Line 202: remove be. Suggest ‘were only bound by Upc2A’

36. Line 203: ‘multiple different conditions’ can be changed to different conditions, or other conditions. Or suggest removing completely after the word fluconazole.

37. Line 205-206: According to the mRNA expression results in Fig1- the mutant Upc2A was not really induced without FLC- except for ERG11. I know that is not global look at genes but the majority of genes analyzed in Fig 1A were NOT induced by the mutant form. So I’m not sure you can call adding fluconazole to that strain an induced condition.

38. The 33 genes that are only bound under FLC treatment are more interesting. And there is no mention of that group.

39. Line 212: again with the use of term inducible- maybe this is just a stressed condition. Drug treated causes stress or mutant protein causes stress. Whereas the 33 genes bound by both Upc2A forms in the FLC-treated condition would love to know what GO terms are in there. Are the enriched genes FLC-specific or general stress induced genes? Could add one other stresser like ROX or other non-ERG disrupting drug. Or heat/chemical stress.

40. Line 212. Put period after the parenthesis and start new sentence with ‘The inducible..’

41. Line 223: What is AR1b/c? were those explained previously?

42. Were the SC SRE element logos generated with the presence of FLC as well? If not, it is not clear that the AR1 logos are relevant to these conditions.

43. Line 224: remove ‘are’

44. Line 227: constant spacing from what?

45. Line 232-233: suggest changing statement to ’… model proposed earlier in which/that suggests Upc2A accumulates inside the nucleus upon ergosterol limitation…’ But isn’t that what you would expect of a transcription factor?

46. Does Upc2 NOT accumulate in the nucleus when erg is not limited? It seems to be inducing expression of CDR1/PDR1 in Fig 1A even without fluconazole or any disruption to ergosterol.

47. How are these p0 mutants created? I would expect Upc2 regulated genes to be upregulated in this strain.

48. Line 244-245: could clarify this sentence by changing ‘along with’ to ‘and’

49. Line 246: Caveat is that this is under a very specific set of conditions that two different strains were compared.

50. Line 247: What are the two classes of promoters? Think this sentence can be removed or adjusted to say ‘Two Upc2A target promoters are shown in Fig 2D’

51. Figure 2: what is the p+ strain? Why are the p0 and p+ strains not listed in the Strain Table?

52. Line 271: ‘union’ could be changed to collection

53. Lines 271 to 287 describe Figure 3. The explanation of this figure is very confusing. The section needs to be rewritten leading the reader through the figure one step at a time.

54. Line 271-274: Sentence is very long. The gene groups being compared in this figure could be described in more detail, using separate sentences.

55. Line 275: I would clarify that the 53 genes required the presence of Upc2A AND correspond with the data for Upc2A binding, to be induced by fluconazole. Rather than just ‘the presence of Upc2A DNA-binding…’ And end the sentence after ‘fluconazole’.

56. Line 275-276: I would also change this sentence to state that there were 274 genes that were found to be bound by Upc2A that do not actually require the presence of Upc2A to be induced by fluconazole.

57. Line 277: The placement of the number in parenthesis probably needs to be moved. Is this number supposed to indicate the number of fluconazole-induced genes that are not dependent on the presence of Upc2A? If so, the number should be moved from proceeding ‘genes’ on Line 277 to proceeding ‘Upc2A’ on line 278.

58. Where is the number 542 coming from? Clearly all genes that are induced by fluconazole in the upc2 deletion strain circle do not require upc2 presence or its promoter binding. So if you are indicating “not dependent on the presence of Upc2A“ as the text says- I’m counting 816 genes. 258+ 284+ 201+ 73. All of the genes in the yellow circle.

59. Line 278: If the genes require the presence of Upc2A but Upc2A does not bind the promoter, what purpose could Upc2A be playing? Could it be that the promoter-binding analyses missed some Upc2A -promoter interactions? Or what are the thoughts on this discrepancy?

60. Line 301: text says the top three enriched categories but then only lists 2- translation and the ribosome.

61. I think the 73 genes on the right side of the Venn that are bound by Upc2 and are NOT overexpressed when Upc2A is present but ARE overexpressed when Upc2 is absent is an interesting group. Could these genes be repressed by Upc2a?

62. Lines 307+: Not sure this part B of figure 3 is cohesive to Part A. part A had nothing to do with Pdr1 regulation. Maybe this data set can be joined with the data from Figure 2 C and D to make a figure correlating Pdr1 and Upc2.

63. Lines 314-315, 318: Keep mentioning WT-cells. Are you comparing the results in the WT strain to results in the Upc2A deletion strain? You don’t mention that in the figure legend or clearly, figure label, and it is not mentioned from lines 307-310. Needs to be clarified in the text and legend and figure label.

64. Line 323: suggest changing ‘gene backgrounds’ to ‘strains’ or ‘in both wild-type and upc2A� cells.’ Can remove ‘irrespective of the presence of Upc2A.

65. Line 348: ‘coordinated’

66. Line 387: reword ‘to those that lacking Upc2A’

67. Line 395: ‘drug-induced’

68. Line 444-446 – Is there a reason you did not show mRNA expression of these genes? The mRNA preparations were used in Fig 6A.

69. Line 514-515: change ‘regulation of ERG gene biosynthesis’ to either ‘regulation of ERG biosynthesis genes’ or ‘regulation of ergosterol biosynthesis’

70. Line 528: Can remove ‘in this membrane’

71. Line 529: “of this membrane compartment”? could it just be – ‘composition of the plasma membrane.’?

72. Line 532: suggest removing ‘the members of’

73. What about FLC treatment? How does pdr1 expression change during flc treatment? And is there a pdr1 ko? Is cdr1 expression altered? Do we know the SRE deletion results in 6C are b/c of upc2a?

74. Line 547: Change the phrasing slightly. ‘constructing the allele’ does not demonstrate that there is no prohibition. But the results of the analysis/functional analysis demonstrate this.

75. Line 548: What would ‘prohibition of hyperactive allele’ look like?

76. Line 549: ‘allele, from which we derived’

77. Line 553-554: What clusters? Mutations?

78. Line 562: shows ‘a’ constitutively

79. Line 991: Add space after (M)

80. Line 992: Strains were ‘grown’ in the presence?

81. 994: with ‘the’ largest number

82. So this means some genes are repressed by UPC2? Which genes are those that are repressed under normal conditions- 7 genes I see.

83. Line 996: Are these peaks and logos WT upc2 after FLC treatment? That should be clarified in the figure legend.

84. Line 1009: Make ‘Venn Diagrams’ singular

85. Line 1009: change ‘union’ to ‘collection’

86. Line 1011: Suggest adding period after ‘fluconazole’ and start a new sentence that describes the 3rd Venn circle.

**Have all data underlying the figures and results presented in the manuscript been provided?**

Reviewer #1: Yes

Reviewer #2: Yes

Reviewer #3: Yes

PLOS authors have the option to publish the peer review history of their article (what does this mean?). If published, this will include your full peer review and any attached files.

Reviewer #1: No

Reviewer #2: No

Reviewer #3: No

---

## [Decision Letter · Decision Letter 1]

10 Sep 2021

Dear Scott,

Thank you very much for submitting your revised Research Article entitled 'The Candida glabrata Upc2A transcription factor is a global regulator of antifungal drug resistance pathways' to PLOS Genetics.  This revised version was reviewed by the same three reviewers who reviewed the original submission.  The reviewers found the manuscript significantly improved. 

The manuscript was fully evaluated at the editorial level and by independent peer reviewers. The reviewers appreciated the attention to an important topic but identified some concerns that we ask you address in a revised manuscript.  Reviewer 1 noted a few areas they felt you would like to address--these seem relatively minor. Reviewer 2 felt that while the manuscript had been reviewed, they felt you could go further in some areas providing additional information, detail, and clarification as outlined in their review in detail. Reviewer 3 found the manuscript improved and had no additional comments or suggestions and felt it was a solid presentation.

Given these review comments we are returning the manuscript to you for one final round of revision and we ask that you take reviewer one and two comments into consideration in finalizing this for acceptance and publication.

We therefore ask you to modify the manuscript according to the review recommendations. Your revisions should address the specific points made by each reviewer.

[LINK]

Please let us know if you have any questions while making these revisions.  We look forward to receiving your revised manuscript and appreciate your interest and support of publication in PLOS Genetics.

Yours sincerely,

Joseph Heitman, MD, PhD

Associate Editor

PLOS Genetics

Gregory P. Copenhaver

Editor-in-Chief

PLOS Genetics

Reviewer's Responses to Questions

**Comments to the Authors:**

Reviewer #1: The authors have done a fantastic job; the presentation now brings out the significance of the findings very effectively.

There are some minor details for the authors' consideration that I may have missed last time:

1. Lines 24-25. I believe that "which" is more appropriate than "that." I would say, "The most commonly used antifungal drugs are the azole compounds, which interfere with biosynthesis of the fungal-specific sterol: ergosterol." I could not easily find the relevant Strunk and White advice, but the principle is summarized well in this posting from Purdue University:

https://owl.purdue.edu/owl/general_writing/grammar/that_vs_which.html

Yes, I know that there are folks who dispute this advice. It's up to you in the end, of course.

2. Line 238 and Figure 3. Just asking: are you sure that you want "WTF" in your publication? Another reviewer mentioned this issue previously and you responded that no other reviewer had mentioned it. But the fact that I missed it last time was just due to distraction, laziness, or any of the seven deadly sins.

3. Line 345. This line looks like a subheading but it is not indicated in bold as the others are.

Reviewer #2: Original Comment:

1) Several interesting observations are made with the ChIP-Seq. There are a large number of promoters (309) that are always bound by Upc2. There are also a large number of promoters that are bound in wild type upon fluconazole treatment and in the gain-of-function mutant (85). Yet this was a small fraction of the total number of promoters bound by Upc2 under FLC treatment. Was this small overlap expected? I would have thought for a true gain-of-function mutation, this overlap would be more substantial. Is there differences in the SRE promoter sequences under these different conditions?

Additional clarification and discussion is needed to explain why out of ~1,000 peaks identified in the ChIP-Seq that were bound by wild-type Upc2A in the presence of fluconazole, only 85 were detected in the gain-of-function mutant. Adding the statement included in the response to the review (or a similar sentiment) in the text itself would highlight this discrepancy to the reader:

“The large number of genes that only show binding in wild-type cells that are FLC-treated was surprising. It is important to note that these represent classes of genes that are not typically thought of as Upc2A-responsive, such as components of the translational machinery, and confirmation of the role of Upc2A in their expression awaits further experimentation.

Original Comment:

2) For the RNA-seq data, the authors find 53 transcripts that require Upc2A for expression and that are bound by the regulator. These represent the direct targets. These should be more clearly described and highlighted in the text. Furthermore, I found it surprising that only 11 genes were significantly upregulated in the Upc2A gain-of-function (GOF) mutant. How do these compare with the 53 transcripts? Overall, my interpretation of this analysis is that Upc2A predominantly governs genes involved in the biosynthesis of ergosterol. The conclusion the Upc2A regulates other diverse sets of genes is likely largely indirect.

Additional clarification is needed for the RNA-Seq analysis. The authors need to clearly articulate that only 53 of the transcripts identified (representing less than 3% of transcripts analyzed) are both bound by Upc2 and induced upon fluconazole treatment in an azole-dependent manner. The other groupings of transcripts described in this section likely represent indirect targets (i.e. the 64 genes that are fluconazole induced in a Upc2A-dependent manner but not bound) or are transcripts that could be induced by Upc2A as well as other transcriptional regulators (i.e. the 274 genes that are fluconazole induced regardless of the presence or absence of Upc2, yet are also bound by Upc2A). A careful revision of this section is needed not only clearly state the conclusions of these findings, but also to walk the reader through the data in a logical manner.

Original Comment:

3) Ergosterol biosynthesis genes (ie ERG11) were shown to be exclusively bound by Upc2A (Fig. 2D) as opposed to being bound by both Upc2A and Pdr1 (ie CDR1). Therefore, it was puzzling in Fig. 5 when the authors introduced a mutant SRE sequence into the promoter regions of CDR1 and PDR1 but not ERG11. I would like to see the impact of mutating the SRE on the expression of a gene that is not also regulated by another key transcriptional regulator. How does mutating the ERG11 SRE impact fluconazole resistance? If there was a reason this was only performed in with ERG1 and not ERG11, this should be explained in text.

Thank you for adding this clarification. While I would still like to see the impact of mutating the SRE in the promoter of a gene ONLY regulated by Upc2A, I appreciate the need for a strong signal and am satisfied with at least including the rationale in the txt.

Original Comment:

4) Similarly, for assessing the role of Upc2A in anaerobic growth, I would like to see how mutating the ERG11 SRE impacts the induction of ERG11 in aerobic versus anaerobic conditions.

No longer applicable due to removal of data.

Original Comment:

5) From what I can tell, the only experiment that specifies number of replicates performed in the analysis is for the western blots. Please specify how many times the other experiments were performed. (Be specific regarding biological or technical analyses). This is especially important for qRT-PCR for which some of the “significant” changes are minimal.

Thank you for adding the details for the qRT-PCR experiments. However, this detail should be included for all other assays as well (spottings, B-galactosidase assay, DNaseI protection assay etc).

Original Comment:

6) Overall the authors should revise the manuscript to ensure language and background information is as clear as possible. For example, line 143-146 should specify that it replaced the native UPC2A and line 191 should explain the p0 cells are C. glabrata without mitochondria. Furthermore, there were several typos such as line 202, take out “be”; line 450, change “early” to “earlier”; and line 549, add “where” in between “from we”.

Thank you for making these editorial corrections. As also pointed out by reviewer 3, additional care should be taken in the final revision to ensure all grammar and spelling are correct, as well as to revise the manuscript for clarity.

Minor Comments:

All have been addressed except:

Original Comment:

The description of the genomic analyses (ChIP-Seq and RNA-seq) was confusing at times. Refining the text to provide systematic and clear Description of the findings would be beneficial.

Insufficient clarification/editorial revision was conducted for ChIP-Seq/RNA-seq sections. While the addition of sub-headings was helpful, this alone is not sufficient to provide the reader with extra clarity for these results. Specifically, for the RNA-seq only a few short sentences were added.

Reviewer #3: None

**Have all data underlying the figures and results presented in the manuscript been provided?**

Reviewer #1: Yes

Reviewer #2: Yes

Reviewer #3: **No: **There are large data sets for Chip-See and RAN seq and I do not know if those data sets have been provided.

PLOS authors have the option to publish the peer review history of their article (what does this mean?). If published, this will include your full peer review and any attached files.

Reviewer #1: No

Reviewer #2: No

Reviewer #3: No

---

## [Editor Report · Decision Letter 2]

22 Sep 2021

Dear Scott,

We are pleased to inform you that your revised manuscript entitled "The Candida glabrata Upc2A transcription factor is a global regulator of antifungal drug resistance pathways" has been editorially accepted for publication in PLOS Genetics. Congratulations!

Yours sincerely,

Joseph Heitman, MD, PhD

Associate Editor

PLOS Genetics

Gregory P. Copenhaver

Editor-in-Chief

PLOS Genetics

Comments from the reviewers (if applicable):

**Data Deposition**

http://datadryad.org/submit?journalID=pgenetics&manu=PGENETICS-D-21-00592R2

**Press Queries**

---

## [Editor Report · Acceptance letter]

27 Sep 2021

PGENETICS-D-21-00592R2 

The Candida glabrata Upc2A transcription factor is a global regulator of antifungal drug resistance pathways 

Dear Dr Moye-Rowley, 

We are pleased to inform you that your manuscript entitled "The Candida glabrata Upc2A transcription factor is a global regulator of antifungal drug resistance pathways" has been formally accepted for publication in PLOS Genetics! Your manuscript is now with our production department and you will be notified of the publication date in due course.

With kind regards,

Amy Kiss

PLOS Genetics

On behalf of:
